# Bifurcation of potential vorticity gradients across the Southern Hemisphere stratospheric polar vortex.

Jonathan Conway[1], Greg Bodeker[1], and Chris Cameron[1]

[1]Bodeker Scientific, 42 Russell Street, Alexandra, 9320, New Zealand

*Correspondence to:* J.P. Conway (jono@bodekerscientific.com)

**Abstract.** The winter-time stratospheric westerly winds circling the Antarctic continent, also known as the Southern Hemisphere polar vortex, create a barrier to mixing of air between middle and high latitudes. This dynamical isolation has important consequences for export of ozone-depleted air from the Antarctic stratosphere to lower latitudes. The prevailing view of this dynamical barrier has been an annulus compromising steep gradients of potential vorticity (PV) that create a single semi-permeable barrier to mixing. Analyses presented here show that this barrier often displays a bifurcated structure where a doubled-walled barrier exists. The bifurcated structure manifests as enhanced gradients of PV at two distinct latitudes - usually on the inside and outside flanks of the region of highest wind speed. Metrics that quantify the bifurcated nature of the vortex have been developed and their variation in space and time has been analysed. At most isentropic levels between 395K and 850K, bifurcation is strongest in mid-winter and decreases dramatically during spring. From August onwards a distinct structure emerges, where elevated bifurcation remains between 475K and 600K, and a mostly single walled barrier occurs at other levels. While bifurcation at a given level evolves from month to month, and does not always persist through a season, inter-annual variations in the strength of bifurcation display coherence across multiple levels in any given month. Accounting for bifurcation allows the region of reduced mixing to be better characterized. These results suggest that improved understanding of cross-vortex mixing requires consideration of the polar vortex not as a single mixing barrier, but as a barrier with internal structure that is likely to manifest as more complex gradients in trace gas concentrations across the vortex barrier region.

## 1 Introduction

The polar vortex is the defining dynamical feature of the stratosphere during winter and early spring. The steep gradients in trace gas concentrations and in dynamical tracers such as potential vorticity (PV) across the vortex boundary region indicate a dynamical barrier to the mixing of air between the mid-latitude and the high latitude stratosphere (e.g. Schoeberl et al., 1992; Hoskins et al., 1985; Plumb and Ko, 1992). In the Southern Hemisphere, during winter and spring, the barrier isolates the cold stratospheric air that develops during the polar night from warmer mid-latitude air. The polar stratospheric clouds that form in this region provide the conditions needed for the heterogenous chemical reactions that create the halogen radicals that go on to deplete ozone (Solomon, 1999 and references therein). In springtime, the barrier prevents ozone depleted air within the Southern Hemisphere polar vortex from mixing with ozone rich air that accumulates in the downward branch of the Brewer-Dobson circulation equatorward of the vortex. By defining the location of the polar vortex in equivalent latitude (EL)

coordinates (Butchart and Remsberg, 1986), the temporal evolution of the size and strength of this dynamical containment of ozone depletion can be assessed (e.g. Bodeker et al., 2002) and periodic mixing of air masses into and out of the vortex can be identified (e.g. Schoeberl and Newman, 1995; Paparella et al., 1997; Ajtić et al., 2004, Manney et al., 2005). Providing a quantitative description of the vortex barrier also allows the representation of the polar vortex in coupled chemistry-climate models to be evaluated (e.g. Struthers et al., 2009).

The centre of the vortex boundary region is most often defined as (i) the EL of the maximum gradient of PV with respect to EL (herein referred to as PV gradient), with the caveat that the true peak is also located in the region of the maximum westerly jet (e.g. Nash et al., 1996), or (ii) the EL of the maximum of the PV gradient times the average wind speed (e.g. Bodeker et al., 2002; Manney et al., 2007). The inner and outer edges of the vortex boundary region can be assessed using the change in the curvature of the vortex definition function (PV gradient or PV gradient times wind speed). Because there is a one-to-one correspondence between PV values and EL (see Figure 1a of Nash et al., 1996), the PV values corresponding vortex boundary region can be used to map the size and location of the vortex in latitude-longitude coordinates. Methods to map the vortex in two dimensions often use climatological PV values corresponding to the vortex boundary (e.g Waugh and Randel, 1999; Manney and Lawrence, 2016) because PV gradients exhibit large temporal variability and because the vortex is sometimes poorly defined. Analyses using climatological PV values can represent inter-annual variations in vortex attributes well, but may not represent the size of the Southern Hemisphere polar vortex well during the spring, as the PV value associated with the strongest PV gradients changes (e.g. Waugh and Randel, 1999). Besides diagnostics using PV gradients, the vortex boundary can also be defined using maximum wind speed in longitudinal bins (e.g. Rummukainen et al., 1994), maximum kinetic energy (e.g. Paparella et al., 1997), concentrations of long-lived trace gases (e.g. McDonald and Smith, 2013), Lagrangian metrics such as Lyapunov exponents or Function M (e.g. Garny et al., 2007; Smith and McDonald, 2014; Serra et al., 2017), the complexity of tracer gradients through effective diffusivity (e.g Haynes and Shuckburgh, 2000), or a combination of different metrics (e.g. Manney and Lawrence, 2016; Lawrence and Manney, 2018). Because of the minimal computational requirements and its basis as a dynamical constraint on mixing, the PV gradient is an attractive method to define the vortex boundary region.

The increased resolution of recent reanalyses (Fujiwara et al., 2017) now allows us to resolve the structure of PV gradients in EL coordinates around the polar vortex in much finer detail than was available when the Nash et al. (1996) definition was proposed. This paper investigates PV gradients during late winter and spring over the altitude range where the highest rates of Southern Hemisphere ozone depletion occur (mid-stratosphere). The aim of this paper is to characterise, in detail, PV gradients across the Southern Hemisphere polar vortex dynamical barrier within the mid-stratosphere, including the vertical and latitudinal structure, seasonal evolution and internal structure of the vortex boundary region.

## 2  Methods

### 2.1  Data sets

The primary data set used in this paper is the ERA-Interim reanalysis (Dee et al., 2011) produced by the European Centre for Medium-Range Weather Forecasting (ECMWF). This reanalysis uses a 4D variational assimilation system to combine

observed climate variables in a dynamically consistent fashion. Zonal and meridional wind speed, and Ertel potential vorticity (PV) data were downloaded on nine potential temperature ($\Theta$) levels (350K, 370K, 395K, 430K, 475K, 530K, 600K, 700K, 850K) from the ECWMF online archive on a regular latitude-longitude grid with a resolution of 0.75° in both dimensions at a 6-hourly timestep for the period 1979-2016. To ensure that features observed in ERA-Interim are not unique to a single

reanalysis, data from the National Centers for Environmental Prediction (NCEP) Climate Forecast System Reanalysis (CFSR; Saha et al., 2010) and its continuation, the Climate Forecast System version 2 (CFSv2; Saha et al., 2014), were also analysed. For simplicity, the combination of these two NCEP data sets are referred to here as CFSR, though we note slight changes in model parameterizations between CFSR and CFSv2 (see Saha et al., 2014). Zonal and meridional wind speed, temperature and Ertel PV fields from CFSR were downloaded on six $\Theta$ levels (350K, 400K, 450K, 550K, 650K, 850K) on a 0.5° regular

latitude-longitude grid at a 6-hourly timestep for the period 1979-2016. The ERA-Interim reanalysis has a higher vertical resolution than CFSR in the lower and middle stratosphere, so we focus on results from this reanalysis, rather than present all the results using both reanalyses.

     To facilitate comparison of PV gradients on different $\Theta$ levels, PV was scaled to have a similar range of values throughout the stratosphere using the scaling factor $\Theta/\Theta_0^{-9/2}$ (Lait, 1994), with a reference level ($\Theta_0$) of 350K. For ease, PV is also

presented here in PVU ($10^6 Km^2kg^{-1}s^{-1}$), referred to as sPVU when PV has been scaled according to Lait (1994). Note this scaling is similar to that of Dunkerton and Delisi (1986), used in other studies (e.g. Manney and Lawrence, 2016), but that the scaled values are of a different magnitude.

## 2.2   Detection of bifurcated PV gradient

Profiles of PV gradient as a function of EL were calculated by first determining the one-to-one relationship between PV and EL

for each time step and vertical level in the reanalyses. Here, EL was calculated for 181 PV values that were determined using percentile steps between the maximum and minimum PV at each time step. Each hemisphere is treated separately and and in the Southern Hemisphere, PV values are multiplied by -1 to ensure a common treatment with Northern Hemisphere. Of the 181 steps, 174 are linearly spaced between the minimum and 99th percentile, while 7 increasingly small percentile steps capture the extreme tail of the PV distribution (percentiles: 99.5, 99.75, 99.875, 99.9375, 99.96875, 99.984375 and 100). Percentile steps

were used to avoid spurious zero PV gradient values that can occur when using equally-spaced PV steps: when no PV data falls between two equally-spaced PV steps, both PV steps are assigned the same EL and a spurious zero PV gradient value occurs. As well as reducing spurious zero-gradient values, percentile steps reduce ringing in the derivative of PV gradient that can occur when using equally-spaced PV steps. Like any other method, using percentile steps to interrogate the PV-EL relationship results a different coordinate mapping from day to day and level to level. While the PV spacing is non-uniform, the resulting

EL spacing is close to uniform as we take an a-priori guess at a regular EL spacing (by assuming all grids points have the same area), rather than choosing equally-spaced PV values to arrive at the same gradients. We have checked that the PV gradients are not dependent on our choice of the percentile method (not shown).

     To characterise the bifurcation of the vortex barrier, an algorithm was developed to detect secondary peaks in profiles of PV gradient as a function of EL as follows (see also Figure 1a):

– The primary peak is identified by the largest value of PV gradient between -40° and -80° equivalent latitude.

– A secondary peak is identified as a local maximum in PV gradient that exceeds a fraction (peak ratio) of the primary peak, that is located a sufficient distance (separation) from the primary peak. In addition, the PV gradient between the two peaks must decrease by a fraction (dip fraction) of the secondary peak in PV gradient.

In order to capture peaks only at times and locations where the vortex is well developed enough to analyse, we introduce a minimum zonal wind speed of +15.2 m/s (Nash et al., 1996) and a requirement for the PV gradient to be significantly above average (c.f. Manney and Lawrence, 2016), for both primary peak and secondary peaks. The minimum PV gradient (sPVU) varies between levels [395K=0.331, 430K=0.375, 475K=0.466, 530K=0.534, 600K=0.552, 700K=0.558, 850K=0.480] and is defined as the mean + 2 x standard deviation of the PV gradient using data between -5° and -80° EL for all time periods from 1979-2016.

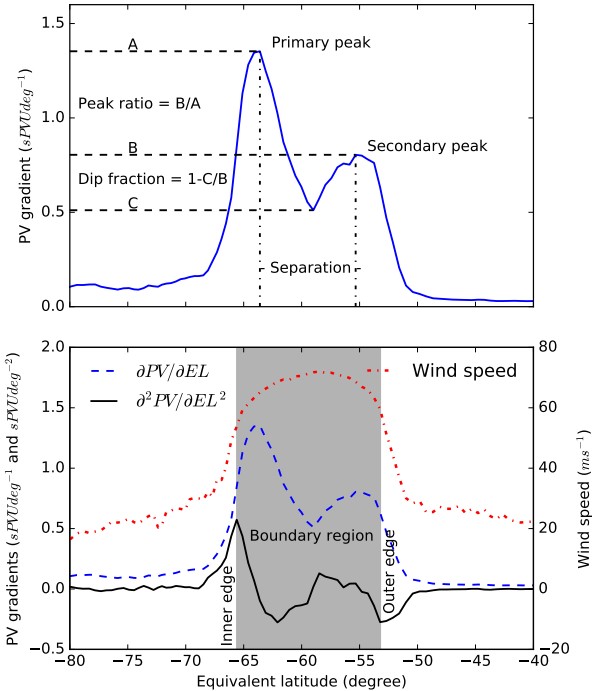

**Figure 1.** Example of (a) the calculation of the peak ratio, dip fraction and separation from a meridional profile of PV gradient ($\partial PV/\partial EL$) and (b) the calculation of inner and outer edges of the vortex boundary region from $\partial^2 PV/\partial EL^2$ ($sPVUdeg^{-2}$), with wind speed ($ms^{-1}$) and $\partial PV/\partial EL$ ($sPVUdeg^{-1}$) also shown for reference. This example is a profile at 530K on 1 September 2007 from ERA-Interim.

A bifurcation index (BI) is derived using combinations of peak ratio and dip fraction that are increasingly restrictive as shown in Table 1. Single peaks are denoted as BI = 0 when the peak ratio is less than 0.5 or the dip fraction is less than 0.05. The threshold values used for determining BI were chosen to capture clearly bifurcated profiles after extensive visual inspect

on PV gradients - hence only peaks with at least half the PV gradient of the main peak are considered bifurcated. In the same way, more than a 5 % dip is required between two peaks, to avoid misclassifying a 'shoulder' on the main peak or noise in the PV gradient as a second peak. The two thresholds increase at the same time so that strong bifurcation requires both a similarly sized secondary peak and a very clear dip between. Without a strong dip, a second peak of similar magnitude more likely represents a wide area of strong PV gradient rather than a distinct peak in its own right. For this reason, the dip fraction has a stronger influence on BI than peak ratio. The analysed values of dip fraction closely follow the threshold values used to define BI, whereas large peak ratios occur in profiles assessed with both small and moderate BI (not shown). Profiles assessed with small to moderate BI can range from two closely matched peaks with only a small interstitial dip, to a much smaller secondary peak on the shoulder of the primary peak. Two choices of separation (2° and 5°) were analysed for all results. Apart from the expected increase in the fraction of bifurcated profiles detected and smaller average BI for 2° separation, the results do not change substantially. The more restrictive 5° separation is used hereafter to highlight the structure of moderate to strong bifurcation.

**Table 1.** Threshold values of peak ratio and dip fraction used to define the bifurcation index (BI).

| Bifurcation Index | Peak Ratio | | Dip Fraction |
|---|---|---|---|
| 0 | < 0.5 | or | < 0.05 |
| 1 | >= 0.5 | and | >= 0.05 |
| 2 | >= 0.55 | and | >= 0.1 |
| 3 | >= 0.6 | and | >= 0.15 |
| 4 | >= 0.65 | and | >= 0.2 |
| 5 | >= 0.7 | and | >= 0.25 |
| 6 | >= 0.75 | and | >= 0.3 |
| 7 | >= 0.8 | and | >= 0.35 |
| 8 | >= 0.85 | and | >= 0.4 |
| 9 | >= 0.9 | and | >= 0.45 |
| 10 | >= 0.95 | and | >= 0.5 |

The inner and outer edges of the vortex boundary region were calculated following Nash (1996); the inner and outer edges are evaluated as the local maximum and minimum in the second derivative of PV with respect to EL around the location of the peak in PV gradient, respectively (Figure 1b). Note that the sign of the second derivative is opposite to the description in Nash et al. (1996) because of the negative sign convention for EL in the Southern Hemisphere. For bifurcated profiles, we nuance the Nash et al. (1996) definition by evaluating the inner edge around the poleward peak and outer edge around the equatorward peak.

# 3 Results

## 3.1 Structure of meridional PV gradients in ERA-Interim

Figure 2 shows the climatological mean (1979-2016) structure of PV gradients across the polar vortex during the winter and spring. Large PV gradients generally align with the region of enhanced wind speed, which is largest at high levels during mid-winter. In August and September, PV gradients intensify in concert with the region of strong wind speed becoming predominately confined to latitudes poleward of -50°. While increased climatological PV gradients occur over a broad region, at lower levels these are nearer the poleward edge of the wind speed jet, while at higher levels these are nearer the equatorward edge of the jet (Figure 2c, d). As spring progresses, the vortex barrier region narrows and the end of October only a narrow region of elevated PV gradients remains.

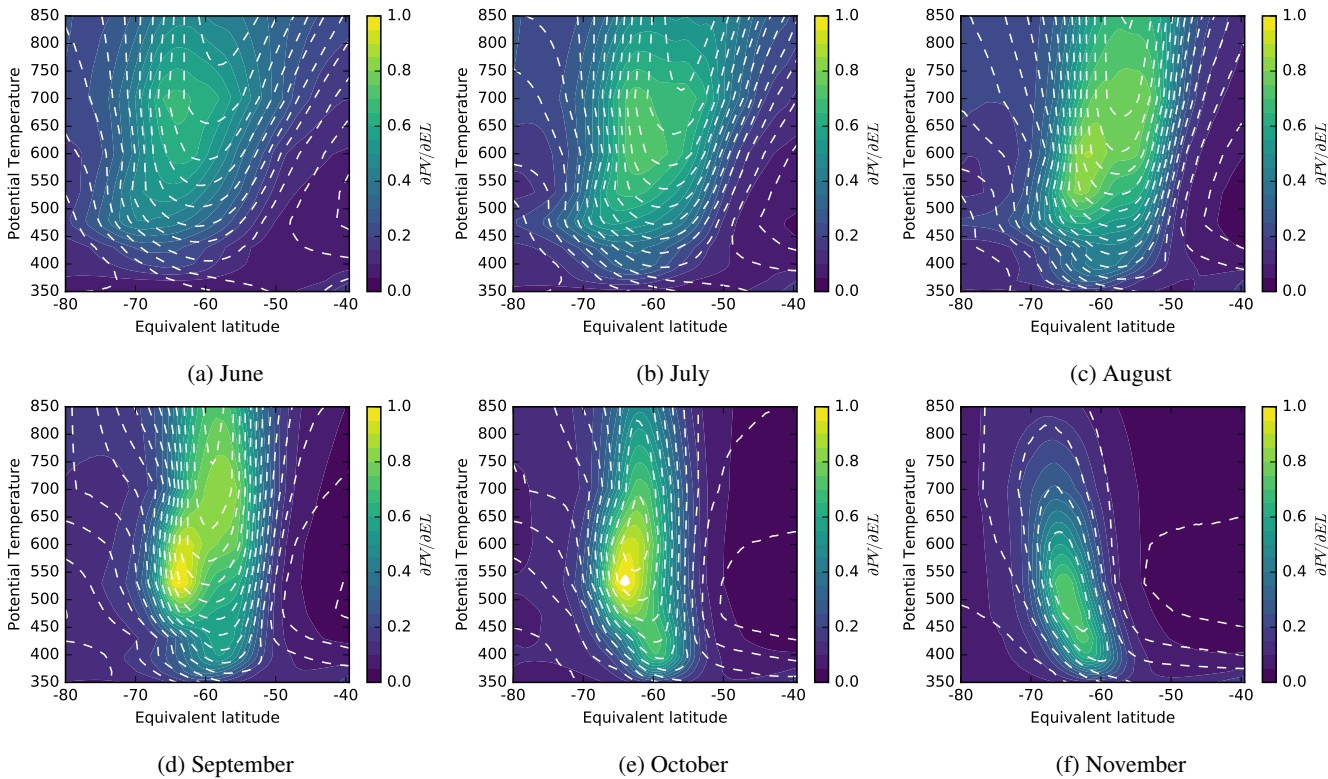

**Figure 2.** Monthly mean PV gradient (colours; $sPVU deg^{-1}$) and wind speed (dashed lines; in 10 ms$^{-1}$ steps) in 0.5° bins at ERA-Interim levels between 350K and 850K, for selected months, 1979-2016.

Considering individual years, the bifurcation of PV gradients with respect to EL becomes more apparent. The characteristic evolution of the bifurcated structure is clearly illustrated during the year 2007 (Figure 3). The PV gradient displays two distinct peaks - one aligned with the transition to steep wind speed gradients on the equatorward side of the jet and a second peak in

a similar position on the poleward flank of the jet. Between these peaks, near the region at the maximum wind speed at mid-levels, are notably reduced PV gradients. The bifurcation of PV gradients is not mirrored in the wind speed, which displays only a single broad region of maximum wind speed. Moderate to high BI occurs at mid-levels during winter and intensifies at 475K during September (Figure 3c).

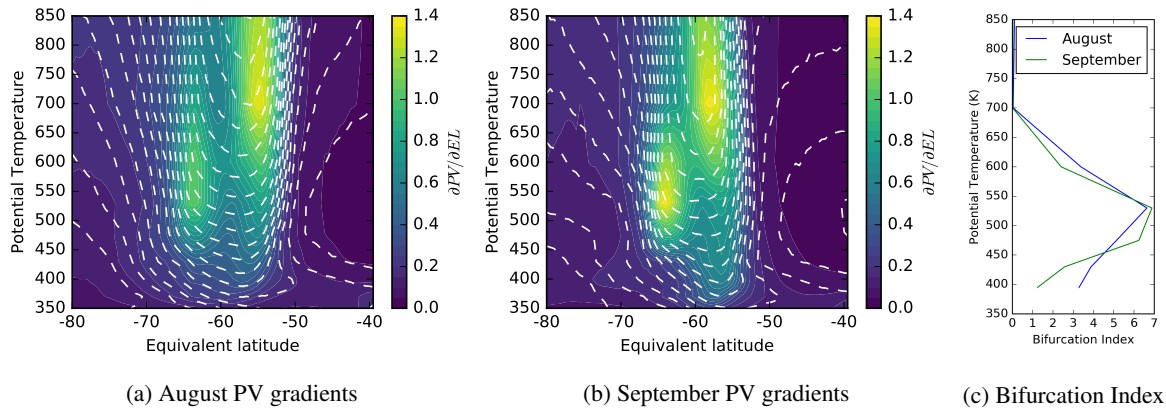

(a) August PV gradients    (b) September PV gradients    (c) Bifurcation Index

**Figure 3.** Monthly mean PV gradient (colours; $sPVUdeg^{-1}$) and wind speed (dashed lines; in 10 ms$^{-1}$ steps) in 0.5° bins during (a) August and (b) September 2007 at ERA-Interim levels between 350K and 850K. Also shown is (c) mean BI at each level for each month.

## 3.2  Bifurcation metrics

Generally, the magnitude of BI and the frequency of bifurcation increases during the winter to peak in July or August (Figure 4a), then decreases during spring. Low BI during October and November indicates that the vortex barrier has only a single peak, or is weakly bifurcated. At mid-levels (475K to 600K), elevated BI remains through September, and, to a lesser extent, October. During September, the barrier is more often than not bifurcated at these levels (Figure 4b). At all levels, the average separation between primary and secondary peaks decreases through the season from around 10° in June to around 5° in October (Figure 4c). During the winter, there is only a slight tendency for secondary peaks to occur equatorward of primary peaks (Figure 4d). From August onwards a more defined spatial structure develops; at mid-levels the secondary peak is most often equatorward of the primary peak. At other levels the secondary peak is more often poleward of the primary peak, though secondary peaks at these levels are not as common or strong as at mid-levels. During September, maximum values of peak ratio occur at 475K and 600K, indicating that two peaks with similar magnitude occur frequently at these levels (Figure 4e). The general decline in dip fraction over the season mirrors the decline in BI, showing its strong influence on BI (Figure 4f).

## 3.3  Location of primary and secondary peaks

The locations of primary and secondary peaks in PV gradient have a well-defined structure during late winter and early spring. At mid-levels (475K to 600K), peaks in PV gradient occur in two preferred locations centred around -63° and -55° (Figure 5),

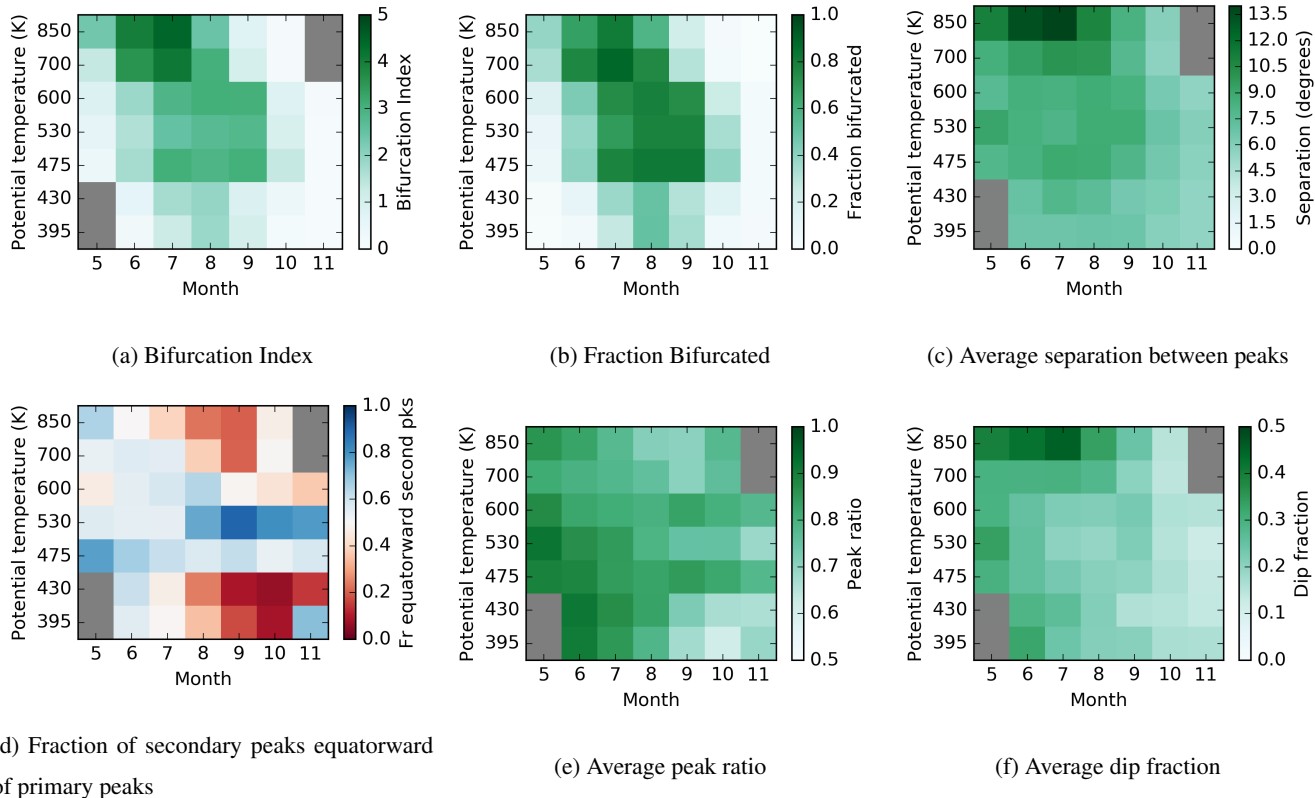

(a) Bifurcation Index

(b) Fraction Bifurcated

(c) Average separation between peaks

(d) Fraction of secondary peaks equatorward of primary peaks

(e) Average peak ratio

(f) Average dip fraction

**Figure 4.** Climatological structure and seasonal evolution of bifurcation metrics derived at ERA-Interim levels between 395K and 850K, 1979-2016. Months/levels with less than 1% of bifurcated profiles are shaded in grey.

with primary peaks being most often in the poleward location, and secondary peaks in the equatorward location. At 700K and 850K, the locations of primary peaks are spread across a wider range of latitude bins, especially in August. At lower levels (395K to 430K), primary peaks are more tightly clustered around -60° and secondary peaks tend to be to poleward of primary peaks.

### 3.4 Inter-annual variability in bifurcation

Large inter-annual variations in BI display some coherence across multiple levels (Figure 6). For example, during August, when bifurcation is common, anomalously low BI occurs simultaneously over multiple levels in a given year (e.g. 1996, 2005). During September, vertically coherent inter-annual variations in BI occur at mid-levels, with anomalies in BI that extend through multiple levels and weak BI above and below. During October, these anomalies persist in some years and are largely confined to 475K to 600K. In November, the single-wall barrier predominates and bifurcation (monthly average BI > 0.75) is only seen in two years (1987 at 475K and 530K, and 2015 at 600K).

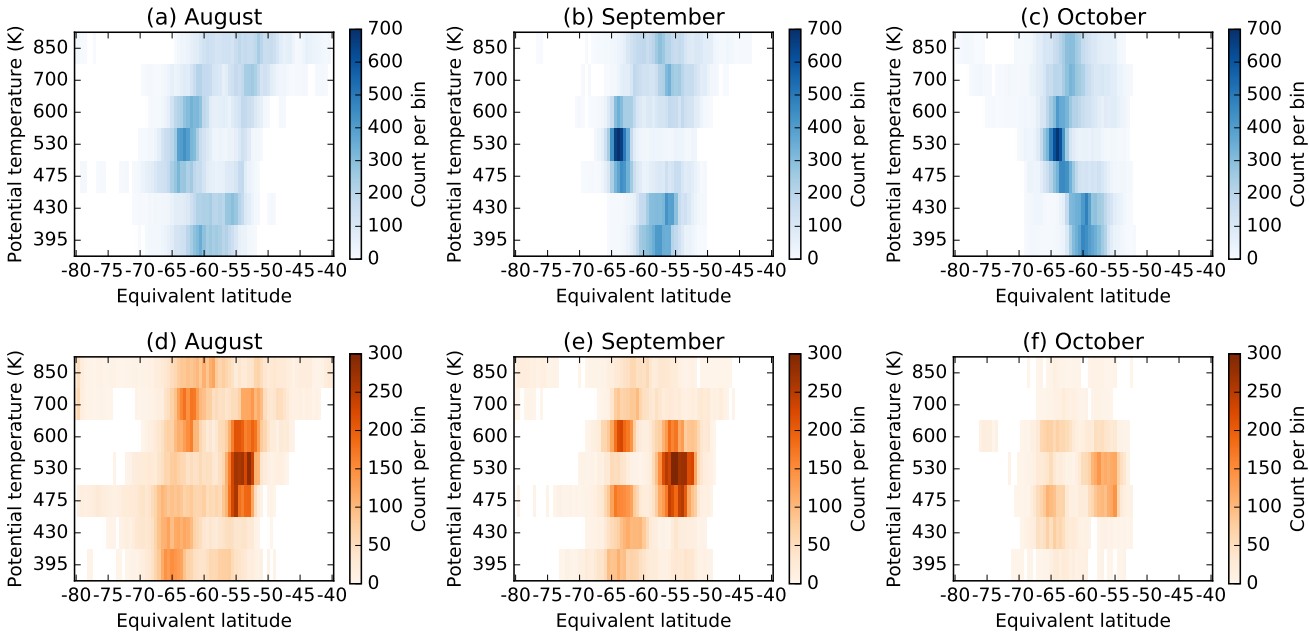

**Figure 5.** Locations of (a-c) primary and (d-f) secondary peaks in PV gradient at ERA-Interim levels between 395K and 850K, 1979-2016. Peaks are selected using a minimum separation of 5° and the colour scale shows the number of time steps that peaks occurred in each 0.5° bin.

## 4  Discussion

### 4.1  Comparison with CFSR

Average BI calculated from CFSR shows a very similar structure to that observed in ERA-Interim (Figure 7). Bifurcation generally decreases during the spring, but remains strong at 550K and 650K. The tendency for secondary peaks to occur equatorward of primary peaks is largely restricted to the 550K layer, with 650K and 450K layers displaying similar structure to those at 700K and 430K in ERA-Interim, respectively. The average locations of primary and secondary peaks are similar, with the primary peaks at 550K being most often in the poleward position (-63°) during September (Figure 8). At 650K and 450K, primary peaks are most often situated at an equatorward location (around -55° to -58°). The occurrence of bifurcation in a second reanalyses gives us confidence that the patterns observed are real phenomena and not artefacts of the production of a particular reanalysis data set.

### 4.2  Impacts of bifurcated PV gradients

The bifurcation of PV gradients across the Southern Hemisphere polar vortex represents a weakening in the rapid meridional decrease in PV that occurs across the vortex barrier. Even though PV gradients weaken between the poleward and equatorward

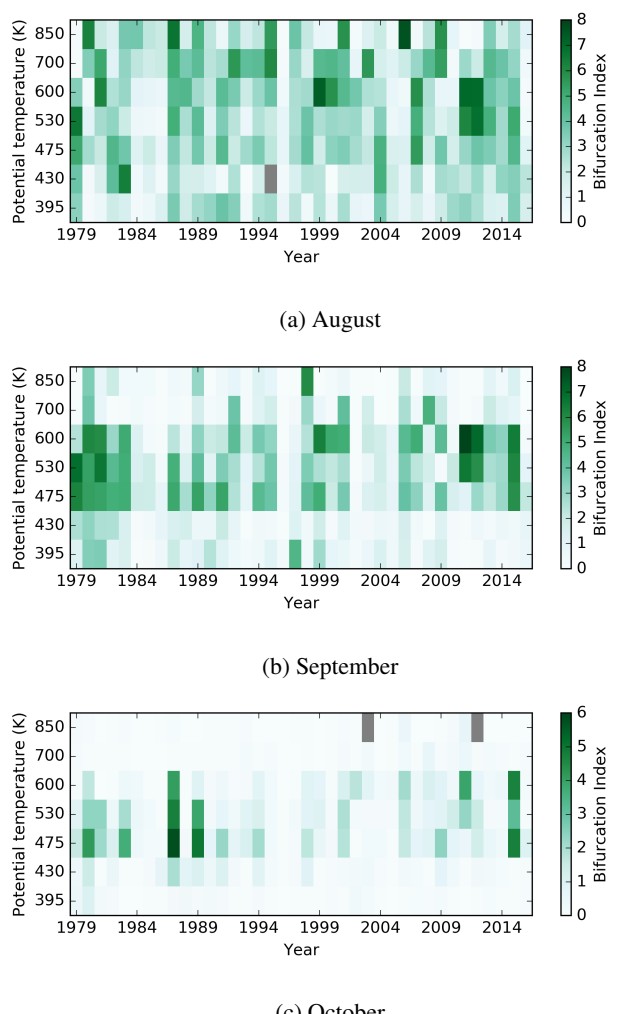

(a) August

(b) September

(c) October

**Figure 6.** Time series of monthly average BI at ERA-Interim levels between 395K and 850K, 1979-2016. Grey colours indicate months with greater than 50% of profiles that do not meet the minimum wind speed and PV gradient criteria.

peaks, they are still substantially larger than those in the mid-latitudes or the vortex interior. To examine the cause of bifurcation in PV gradients, we decomposed PV into its two components, absolute vorticity and static stability ($\partial\theta/\partial p \; [KPa^{-1}]$) , then calculated gradients of these components with respect to EL. Absolute vorticity was calculated from relative vorticity fields downloaded from the ECMWF online archive and latitude at each grid point. Static stability was back-calculated from PV and absolute vorticity at each grid point using the equation for Ertel PV.

The gradients of vorticity and static stability at 530K (Figure 9) reveal the most rapid decreases in PV occur at the points where relative (and absolute) vorticity also display the most rapid decrease - in the transition from the positive vorticity associated with increasing wind speed on the outer flank of the vortex, and to the large negative vorticity associated with decreasing

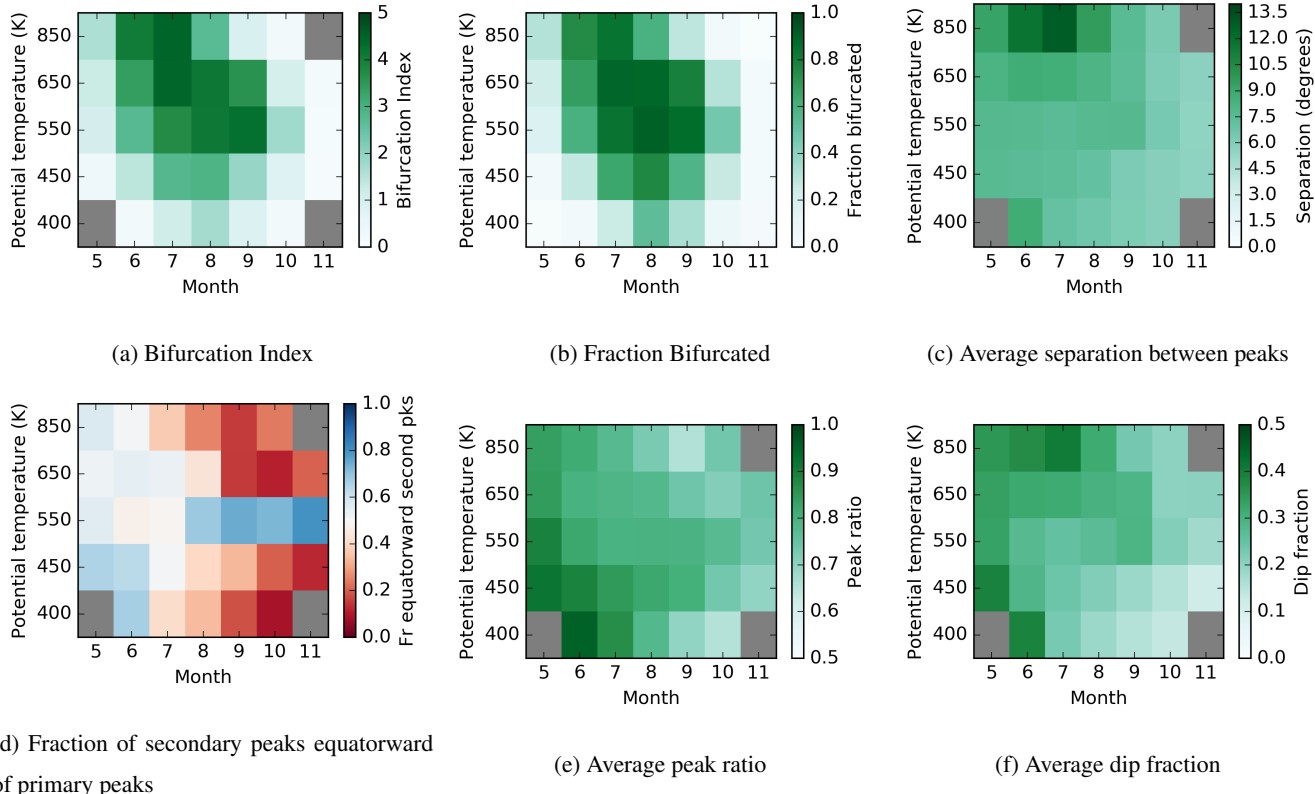

(a) Bifurcation Index

(b) Fraction Bifurcated

(c) Average separation between peaks

(d) Fraction of secondary peaks equatorward of primary peaks

(e) Average peak ratio

(f) Average dip fraction

**Figure 7.** Climatological structure and seasonal evolution of bifurcation metrics at CFSR levels between 400K and 850K, 1979-2016. Months/levels with less than 1% of bifurcated profiles are shaded in grey.

wind speed on the inner flank of the vortex. In addition, the gradient of static stability is greater on the inside flank of the vortex. Thus, bifurcation of PV gradients appears to be caused by the separation of two distinct zones of rapidly decreasing relative vorticity in concert with stronger gradients of static stability on the inner boundary of the vortex. During periods where the barrier is narrow, the two zones of enhanced vorticity gradient coalesce and only a single peak in PV gradient remains.

5      To extend this analysis further, we construct latitude-height composites of gradients in each component; for 1 July from five years with strong bifurcation at upper levels (700 and 800K) and for 1 August and 1 September from five years with strong bifurcation at mid levels (475 to 600K) (Figure 10). The composite analysis reinforces that bifurcation of PV gradients during August and September is associated with elevated vorticity gradients in two locations along with increased static stability gradients at the poleward location. A clear distinction in the location of maximum vorticity gradients is seen during bifurcation

10    at mid-levels during August and September. During years with strong bifurcation at upper levels during July, the location of maximum absolute vorticity and pattern of static stability is less clear, suggesting that the location of bifurcated peaks is more dispersed, and/or the mechanism causing bifurcation at upper levels is more complex.

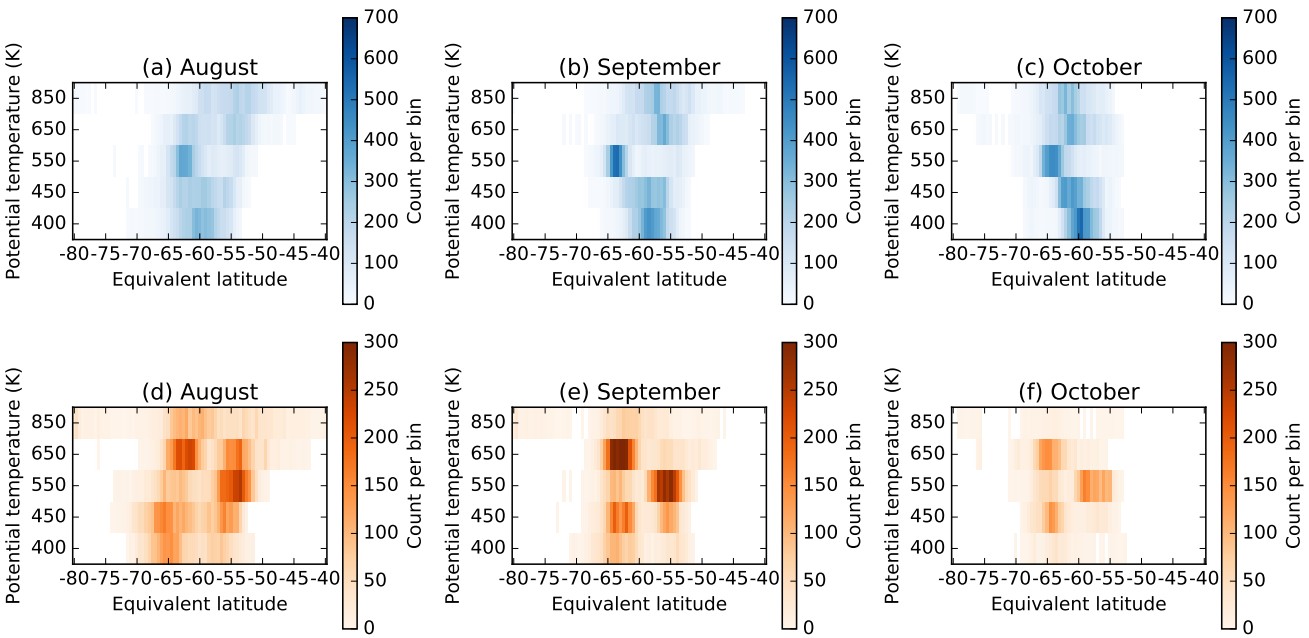

**Figure 8.** Locations of (a-c) primary and (d-f) secondary peaks in PV gradient at CFSR levels between 400K and 850K, 1979-2016. Peaks are selected using a minimum separation of 5° and the colour scale shows the number of time steps that peaks occurred in each 0.5° bin.

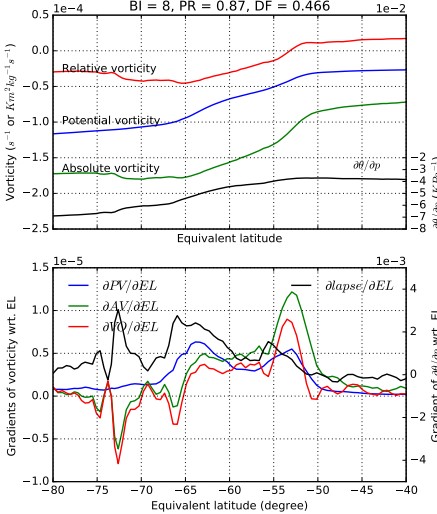

**Figure 9.** Example meridional profile of (a) (LH axis) absolute (AV) and relative vorticity (VO) ($s^{-1}$), PV ($Km^2kg^{-1}s^{-1}$), and (RH axis) static stability ($KPa^{-1}$), and (b) gradients with respect to EL ($\partial PV/\partial EL$ ($sPVUdeg^{-1}$)). All lines are averages at EL calculated from PV. This example is from 530K on 1 September 2011 in ERA-Interim.

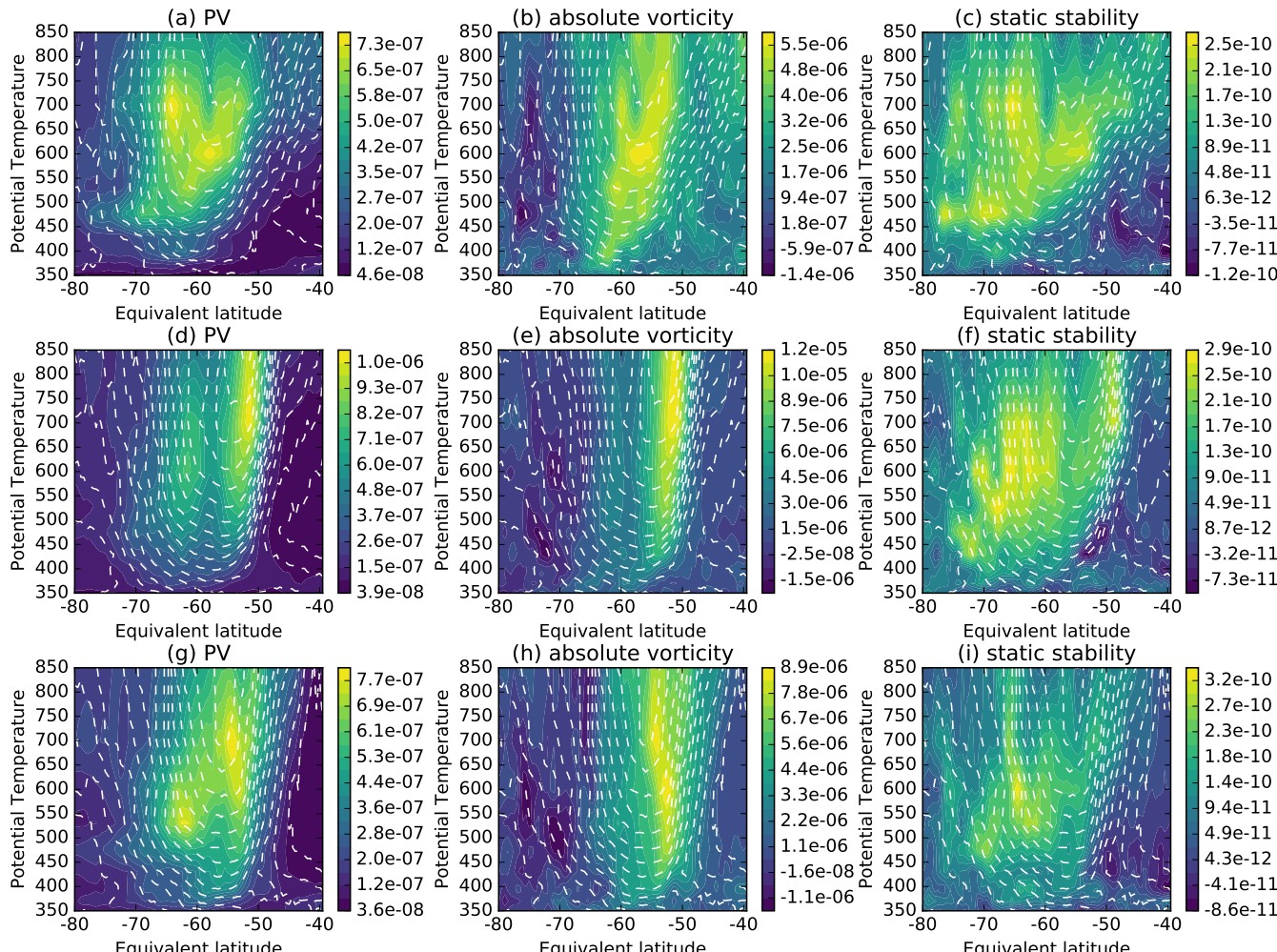

**Figure 10.** Gradients of PV, absolute vorticity and static stability with respect to EL for composites of years with elevated BI in selected months (a-c) July [1983,1986,1989,1991,2005], (d-f) August [1979,2007,2011,2012,2015] and (g-i) September [1980,1981,2011,2012,2015]. See text for further explanation of method used to select years.

The weakening of PV gradients within the vortex barrier during periods of bifurcation may increase mixing within the vortex boundary region, while still reducing the transport of material through the barrier as a whole. This could lead to more complex variations of trace gas concentrations across the barrier. With a single (narrower) barrier, there is less efficient mixing of air and steeper gradients in trace gas concentrations within the vortex barrier region would be expected.

5   Increased PV gradients observed at mid-levels during spring, on the inner flank of the vortex barrier (Figure 5b), align with the outer limits of regions of enhanced ozone loss (Lee et al., 2001; Bodeker et al., 2002). This provides motivation for updated analyses of trends in vortex attributes, such as BI, to provide context for recent reports of trends in ozone concentrations

observed during September (e.g. Solomon et al., 2016). Further work should investigate the dynamical links between ozone loss and PV gradients on the inner boundary of the vortex at mid-levels.

## 4.3 Redefining a vortex barrier region

If a single vortex 'edge' is defined according to Nash et al. (1996), bifurcation can lead to spurious changes in the location
of the vortex boundary region over time. In cases where the PV gradient has two peaks of similar magnitude, the location of the 'edge' will flip-flop from one location to the other over the course of a few days, while clearly the vortex has not rapidly changed in size (Figure 11). This challenges interpretations of the vortex 'edge' that define a single location at the maximum PV gradient. By using the bifurcation algorithm described above to detect secondary peaks in the PV gradient, a more realistic and robust analysis of the location of vortex barrier is made possible.

The Nash et al. (1996) definition of the vortex boundary region is also challenged by the bifurcated structure. The vortex boundary cannot be simply thought of as the region between the local maximum and local minimum in the second derivative of PV with respect to EL around the location of the primary peak in PV gradient. The presence of secondary peaks in PV gradient inside or outside the primary peak highlights that region of limited mixing is, in fact, much wider than that analysed around only the primary peak.

When the bifurcation is taken into account, the width of the vortex boundary region becomes much larger than that analysed with the Nash et al. (1996) definition (Figure 12). During August and September, a clear separation between inner and outer edges becomes apparent at mid-levels. The inner edge is predominately around -65°, with the outer edge at around -52° to -54°. At times, the outer edge is poleward of -60°, but the inner edge is rarely equatorward of -60°. At lower levels the boundary region is usually confined to a small area around the primary peak at -60°, while at higher levels, the location and width of the
boundary region are much more variable. As spring progresses the boundary region coalesces at all levels.

This study suggests there is a need to redefine the location of the vortex barrier as a boundary region in which mixing is reduced below some critical threshold, rather than as a single 'edge'. This definition would demarcate only the inner and outer edges of the vortex boundary region, and would not be dependent on defining a single location for the vortex edge. If a single EL or PV value is needed to demarcate the vortex, then the average EL of the boundary region could still be calculated, but this
value would be more robust against bifurcation than a single 'edge'. The extension of the Nash et al. (1996) methodology shown here shows promise. Defining threshold values of PV gradient may also be a useful way to define the barrier region, though a formal comparison with other metrics of isentropic mixing (e.g. effective diffusivity, Lyapunov exponents) is warranted. This comparison would allow complementary values of each metric to demarcate the inner and outer edges to be inferred.

## 5 Conclusions

The strength of the dynamical isolation of high-latitude air caused by the Southern Hemisphere polar vortex has been analysed using gradients of potential vorticity along equivalent latitude coordinates for seven isentropic levels between 395K and 850K in the ERA-Interim reanalysis for the years 1979-2016. Analyses presented here show that this barrier often displays a bifurcated

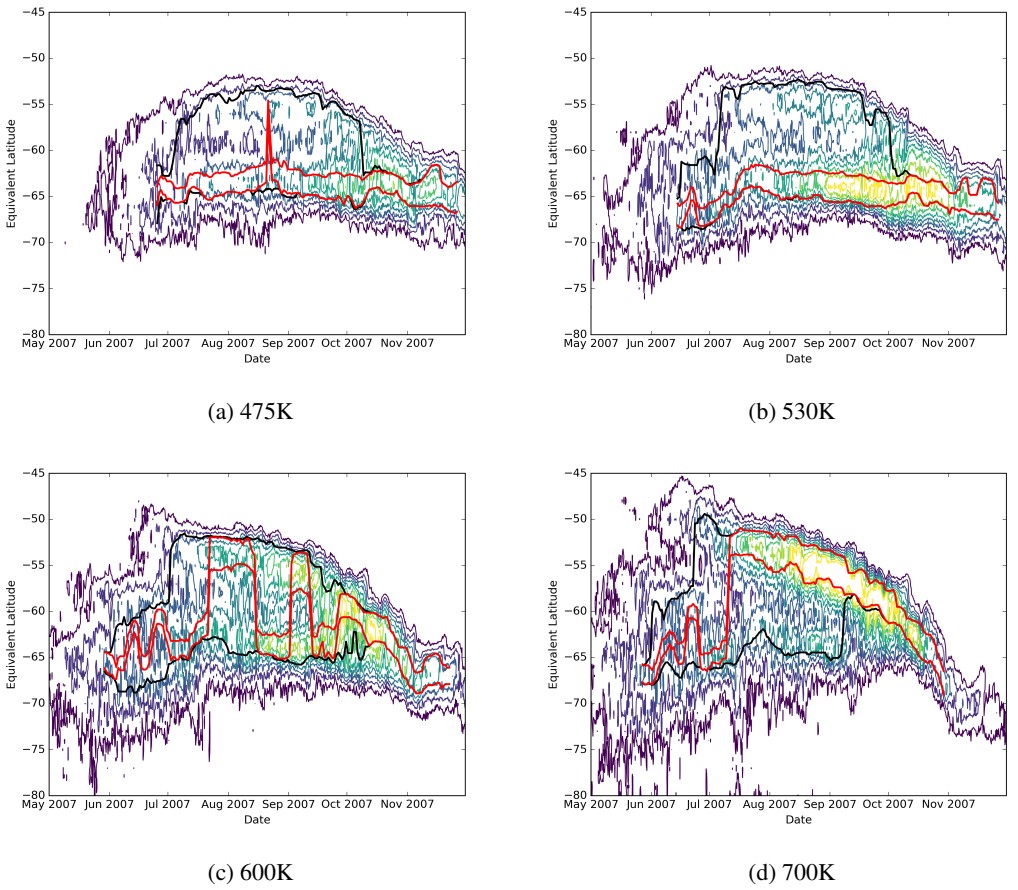

(a) 475K

(b) 530K

(c) 600K

(d) 700K

**Figure 11.** Example showing evolution of PV gradient (coloured contours) and 7-day running average of inner and outer edges diagnosed using (red) the Nash et al. (1996) algorithm or (black) the BI algorithm presented here, during 2007 for four ERA interim heights. Where the black line is not visible, it is in the same position as the red line. Contours of PV gradient are given in 0.15 sPVU steps from 0.3 sPVU to 1.5 sPVU.

structure where, rather than a single barrier to meridional mixing, a doubled-walled barrier exists. The bifurcated structure manifests as enhanced gradients of potential vorticity at two distinct equivalent latitudes - usually the inside and outside flanks of the region of with enhanced wind speed.

Metrics quantifying the bifurcated nature of the vortex barrier have been developed and their variation in space and time has been analysed. While the height-latitude structure differs from year to year, bifurcation appears to be a persistent feature of the vortex barrier region. It was found that at most isentropic levels between 395K and 850K, bifurcation is strongest in winter and reduces dramatically during spring as the barrier becomes more defined. From August onwards, a distinct structure emerges, with stronger bifurcation at mid-levels (475K to 600K) and a mostly single-walled barrier at other levels. During September, peaks in PV gradient show a strong preference for two locations around -63° and -55° and primary peaks at 530K are most

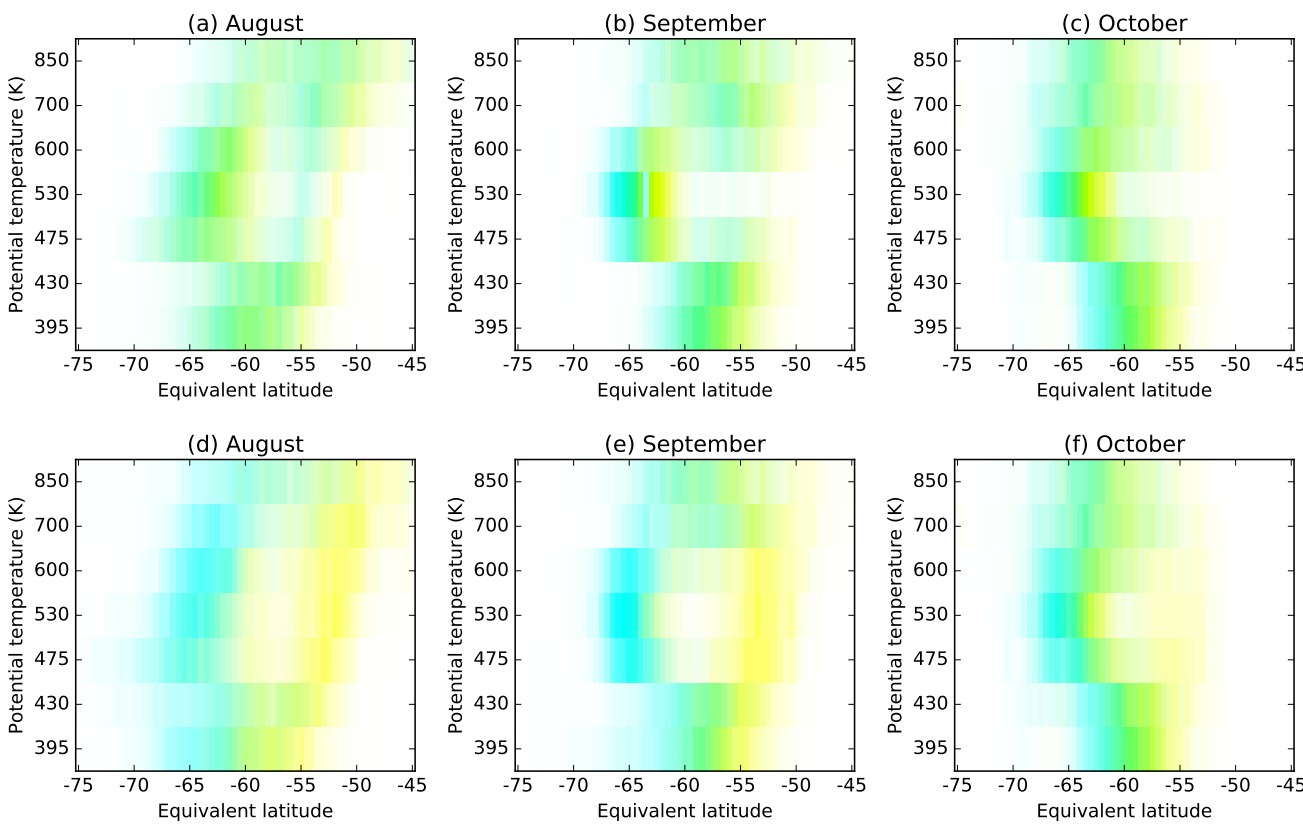

**Figure 12.** Location of inner (cyan) and outer (yellow) edges of the vortex boundary region at ERA-Interim levels between 395K and 850K, 1979-2016, defined using (top row) the criteria of Nash et al. (1996) and (bottom row) accounting for secondary peaks in PV gradient using the bifurcation algorithm presented here. The intensity of cyan and yellow represents the frequency of inner and outer boundaries in each 0.5° bin, respectively, with green colours indicating a mix of both inner and outer boundaries at different times. The frequency per bin is scaled with respect to a maximum count of 600 per bin.

often at the poleward location. While the strength of the bifurcation at a given level evolves from month to month and does not always persist through a season, inter-annual variations display coherence across multiple levels in any given month. Analysis of a second reanalysis, CFSR, shows very similar patterns. The detection of secondary peaks in PV gradient permits a more realistic view of the true region of weak mixing associated with the polar vortex.

5     The results presented above suggest that improved understanding of cross-vortex mixing requires consideration of the polar vortex not as a single mixing barrier, but as a barrier with internal structure that is likely to manifest as more complex gradients in trace gas concentrations across the vortex boundary region. More work is needed to quantify relationships between PV gradients and other metrics of mixing (effective diffusivity, Lyapunov exponents) as well as identify the interactions between

the structure of PV gradients, ozone and temperature. Analyses of trends in bifurcation should also be pursued to provide context to observed trends in ozone concentrations during springtime.

*Data availability.* The ERA-Interim data set used in this paper is available from the ECMWF online archives: http://apps.ecmwf.int/datasets. The CFSR and CFSv2 data sets are available from https://rda.ucar.edu/datasets/ds093.0/ and https://rda.ucar.edu/datasets/ds094.0/, respectively.

*Author contributions.* JC developed and performed the analyses, and prepared the manuscript with contributions from all co-authors.

*Competing interests.* The authors declare that they have no conflict of interest.

*Acknowledgements.* This research was funded by a Royal Society of New Zealand Marsden Fund research grant titled "The permeability of the Antarctic vortex", contract number BDS1401.

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
