# Peer review of "Bifurcation of potential vorticity gradients across the Southern Hemisphere stratospheric polar vortex."

_Atmospheric Chemistry and Physics, 2017_

## Short Comment (SC1) · 26 Nov 2017

Comments on: "**Bifurcation of potential vorticity gradients across the Southern Hemisphere stratospheric polar vortex.**"
written by J. Conway, G. Bodeker, and C. Cameron
for *Atmospheric Chemistry and Physics*

Comments by: Zachary D. Lawrence (zachary.lawrence@student.nmt.edu)

In this paper, Conway et al. use potential vorticity and other data from the ERA-Interim and CFSR reanalyses to highlight a common feature of the southern hemisphere (SH) stratospheric polar vortex: a double peak in gradients of potential vorticity with respect to equivalent latitude (EqL). The authors describe a method to detect this bifurcation by first locating the maximum PV gradient between -80 and -40 EqL (the primary peak), and seeking the next local maximum having a sufficient peak ratio, peak separation, and in-between dip fraction from the primary peak (the secondary peak). They then quantify the bifurcation in the PV gradients by deriving a bifurcation index (BI) using thresholds of the peak ratio and dip fraction diagnostics. Conway et al. use these diagnostics to show occurrence frequencies in monthly and yearly composites that highlight when, where (in potential temperature and/or EqL), and to what magnitude these bifurcated structures arise. They also give a brief examination on how these bifurcations arise in the PV gradients with illustrations of EqL profiles of relative/absolute vorticity and vertical potential temperature gradients, and discuss the need to treat the SH vortex edge/barrier region more carefully.

The paper is generally well-written and clear, and I particularly appreciate its brevity. The subject matter of the SH PV gradient bifurcation alone is significant, and should be of interest to many who study stratospheric dynamics and transport. The detection algorithm and bifurcation diagnostics are also evidently useful. Thus, the paper is an important scientific contribution, and should be published in some form. However, I do think that the paper in its initial submission form suffers somewhat from a lack of references and discussion, as well as some minor organizational issues. Below I discuss my concerns and questions in more detail:

**Comments and Questions**
**1.** The introduction is rather short. While it is nice to have it be concise, the introduction in its present form does not discuss any previous examinations of the SH vortex boundary region. I do not know of any papers that show or mention the bifurcation of PV gradients, but there are many studies (some of which are already cited in the paper) that have discussed and shown that the SH vortex boundary region

is often quite wide, especially relative to the NH vortex. Many of these studies have used more complicated diagnostics than PV gradients (e.g., effective diffusivity) to show this, so mentioning and briefly discussing such papers would reinforce the following statement in the intro *"Because of the minimal computational requirements, the PV gradient is an attractive method to define the vortex boundary region."* Some potential relevant references are already cited in the paper; for example, see figures and related discussion on: Figures 3 – 5 in Paparella (1997) show trajectories of balloons trapped inside the SH polar vortex core and polar vortex boundary; Figures 1 & 2 in Lee et al. (2001) show the width of the vortex boundary region in effective diffusivity during 1996; and plates 1 – 4 in Haynes and Shuckburgh (2000) show monthly means of effective diffusivity that can allow comparison of the widths of the Arctic vs Antarctic vortex boundaries. Other more recent studies that may or may not be relevant to look at include (but certainly are not limited to): de la Cámara et al., JAS, 2012; Abalos et al., QJRMS, 2016; and Curbelo et al., NPG, 2017.

**2.** Tied to 1, the results shown in the current paper could be made more impactful in both the introduction and conclusions by being more clear about the "magnitude" of what is shown. In my opinion, the paper sells itself a bit short. More specifically, I mean that many previous studies (including some of the ones discussed in comment #1) only show results relevant for a single dataset for one to a few winters; this paper shows results relevant for two reanalyses for years from 1979 to roughly the present. I think that is worthy of being highlighted as one of the strong points of the paper.

**3.** This is very minor, but in section 2.1, the years used for ERA-Interim and CFSR should be listed.

**4.** Figure 1 and Table 1 are helpful, but it would be nice if a new figure was included showing EqL profiles of PV gradients for different values of the BI. For example, a 2 x 3 figure with panels showing (along with dates, isentropic levels, peak ratios, and dip fractions listed) representative cases with BI = 0, 2, 4, 6, 8, and 10. I think this would give the reader a better sense of the connection between BI and the geometry of the PV gradient profiles, as well as enhance understanding and the significance of the other figures in the paper.

**5.** In Section 3.1 and Figure 2, is it really accurate to say that there are two regions of enhanced PV gradients? Since there is no clear dip, it looks as if there is only one broad region of enhanced gradients in all cases, even though the location of the climatological maximum gradients moves "equatorward" with height in August & September. This pattern seems to be clearly formed from averaging many

cases with bifurcated PV gradients having peaks in different EqL locations, which would strengthen the idea that the PV gradient bifurcation needs to be examined on a year-to-year basis to really see and understand it (i.e., as is shown in Figure 3).

**6.** In Figure 3, I would be curious to see whether the bifurcated structure also shows up in "regular" zonal means of meridional PV gradients (i.e., with respect to "regular" latitude), since the SH flow and PV distributions are usually relatively close to zonally symmetric. If the bifurcated structure also exists in the zonal mean picture, this could be an interesting result with further implications (e.g., for wave guiding/propagation).

**7.** The comparison with CFSR in section 4.1 is so short that it almost seems unnecessary. First of all, why not show all the same panels for CFSR in Figures 7 and 8 as those in Figures 4 and 5? And since section 4.1 is so short, why not fold the CFSR results from Figures 7 and 8 into Figures 4 and 5 (and maybe even Figure 6) and discuss the comparisons in sections 3.3 and 3.4 directly? Figures 4 and 5 would obviously increase in size (and more labels would be necessary), but there would be an advantage to having the results from the two reanalyses side-by-side for easier comparison, and less figures overall. This would also reinforce one of the strengths of the paper (see my comment #2).

**8.** Figure 9 and its discussion in Section 4.2 would be a bit more useful/significant if Figure 9 showed composites of some form, so that readers would know the discussion is more broadly relevant than to a single day of data. I do realize, however, that this may not be easy or possible to do since the locations of peaks in PV gradients (and hence the geometry of the PV and relative/absolute vorticity profiles) will vary from day to day – but it is something that might be worth a try.

**9.** Also in Figure 9 – how were the fields of relative/absolute vorticity obtained? Were they derived from the potential vorticity and temperature fields, or the wind components, or were they downloaded?

**10.** The following is just an idea I had, but something the authors might consider: The current organization of the paper is fine, but since the paper is largely focused on presenting the methods and diagnostics, there is an alternative organization I see that might flow better for readers (and would only require minimal work). Arguably the bifurcation detection methods and diagnostics are part of the results of the paper. Thus, section 2 could be changed to a "Data and Background" section wherein section 2.1 would remain the same, section 2.2 would discuss how the EqL profiles of PV and PV

gradients are obtained, but a new section 2.3 would be inserted (the background) that included Figures 2 and 3 (which would become Figures 1 and 2) and their discussion. The BI panel in Figure 3 would have to be changed to something like "average peak separation in degrees EqL" since BI would not be introduced yet. Then in section 3, the description of the bifurcation detection and diagnostics (and the current Figure 1) would be included in section 3.1 with a title like "PV gradient bifurcation detection and diagnostics." This organization would lead readers from the climatologies (which smear out the bifurcated gradients), to a single SH winter that illustrates the phenomenon, to the detection methods and bifurcation diagnostics, and finally to the results/figures that directly show the BI and other diagnostics (which, with this organization, would have just been defined). Again, this is just an idea; the authors should feel free to flatly reject it!

**References**

Abalos, M., B. Legras, and E. Shuckburgh: Interannual variability in effective diffusivity in the upper troposphere/lower stratosphere from reanalysis data. Q. J. R. Meteorol. Soc., 142: 1847–1861. doi:10.1002/qj.2779, 2016.

de la Cámara, A., A. M. Mancho, K. Ide, E. Serrano, and C. R. Mechoso: Routes of Transport across the Antarctic Polar Vortex in the Southern Spring, J. Atmos. Sci., 69:2, 741-752, https://doi.org/10.1175/JAS-D-11-0142.1, 2012.

Curbelo, J., V. J. García-Garrido, C. R. Mechoso, A. M. Mancho, S. Wiggins, and C. Niang: Insights into the three-dimensional Lagrangian geometry of the Antarctic polar vortex, Nonlin. Processes Geophys., 24, 379-392, https://doi.org/10.5194/npg-24-379-2017, 2017.

---

## Referee Comment (RC1) · G. Manney (Referee) · 28 Nov 2017

**Review of**
Conway et al.
"Bifurcation of potential vorticity gradients across the Southern Hemisphere
stratospheric polar vortex",

(Gloria L Manney, manney@nwra.com)

**Recommendation:** Should be suitable for publication in ACP with some modest revisions and clarifications.

**General Comments:**
This paper uses ERA-Interim reanalysis data to describe the climatological bifurcation of potential vorticity gradients in the Antarctic vortex edge region. Overall, the analysis is sound and the paper is well-written. To my knowledge, a detailed description/climatology of the bifurcation of PV gradients in the Antarctic vortex edge region has not been published, and the material will thus be of interest to ACP readers. In its current form, the paper is somewhat lacking in referencing previous work related to vortex edge definitions and mixing barriers, and in explaining the rationale behind the definitions of bifurcation index and other threshold choices used. The introduction, in particular, doesn't provide a clear picture of the relevant background and previous work. Once these issues, and some other specific (minor) points listed below, have been addressed, the paper will be appropriate for publication in ACP.

*Note 1:* Obviously, Zachary Lawrence and I work together. However, we were each notified about the paper separately and each read the paper and wrote up our comments that are posted here independently.

*Note 2:* I am not suggesting that you cite every one of the numerous Manney et al. papers mentioned in this review; it is inevitable that I am most familiar with those I wrote, and in some cases they may lead you to more appropriate citations; in a couple of cases they are just mentioned as "of interest".

**Specific Comments (in order of appearance in the paper):**

*Introduction:*

Overall, the authors should make clear that much of this discussion (and which parts of it) applies to the Antarctic polar vortex only; many of the statements made are either not applicable to the Arctic vortex, or would need to be qualified (e.g., the discussion of ozone loss and dispersal from the vortex) to describe the Arctic; this is fine -- the paper focuses on the Antarctic -- but should be said explicitly when there are statements that apply only to the Antarctic. In addition, some statements are rather vague, may not be quite accurate, and/or could use more appropriate citations. Specifically:

- Page 1, lines 20--21: "...cold air formed in the polar night…" -- "formed" doesn't seem quite accurate.
- Page 1, line 19: Schoeberl et al (1992) would be a good reference to add to those here; also please use "e.g.," ahead of the reference list in cases like this where there are many papers relevant to the statement.
- Page 1, line 22: Add "e.g.," before or "and references therein" after "(Solomon, 1999)"
- Page 1, lines 23--24: Please emphasize that this applies to the Antarctic vortex
- Page 2, lines 1--2: There are many, some earlier, references that could be used here for both statements (I'm sure that is also the case for the statement on line 3, though I am not well-read on such CCM studies). At the very least, "e.g.," or "and references therein" should be used. Some other useful papers to cite (including only ones specifically relevant to the SH) might be:
    - Schoeberl et al, 1995 (JGR)
    - Lary et al, 1996 (QJ)
    - Manney et al, 1999 (JGR), 2005 (JAS), 2005 (GRL)
    - Ajtić et al, 2004 (JGR)
- Page 2, lines 4--18: There are some vortex edge definition methods that are not mentioned here, as well as relevant papers It would be good to cite to briefly discuss other ways the vortex edge or boundary region can be defined besides those you already mention. Also, the statement that "Bodeker et al. (2002) formalised this method" is not accurate: Determining the vortex boundary using the maximum in windspeed x PV gradient is not the same as determining it by the maximum PV gradients but requiring that maximum to be in proximity to the windspeed maximum (see, e.g., the brief discussion in Manney et al., 2007, who implemented and compared the two methods, starting with code provided by Eric Nash for the method described in Nash et al). It would also be good to mention some other papers that discuss different ways of defining the vortex edge, especially some of the early papers that use PV gradients (eg, Trounday et al, 1995) and/or use other methods and compare with PV gradient methods (eg, Paparella et al, 1997, JGR, already cited here but the comparisons are not mentioned; Rummukainen et al, 1994, Ann. Geophys.) Smith and McDonald (2014, JGR) used the "function $M$" to look at the strength of the Antarctic vortex and the width (area) of the vortex boundary region. The authors may be interested (though certainly not obliged to cite a paper that will only be available online sometime in the next few days after I'm writing this review) in the vortex edge method in Lawrence and Manney (2017, JGR, in press), which also gives further discussion of various vortex edge definition methods and literature.
- Page 2, line 12: Add "e.g.," before "Manney and Lawrence, 2016" (I am sure others have done this as well; our two Manney et al (2015) papers are ones where I know this was done).
- Page 2, lines 11--18: May be worth noting that Manney and Lawrence (2016) compared their climatological selection of vortex edge PV values to daily PV gradients, effective diffusivity, the function $M$, and trace gas gradients.

*Section 2.1:*

It would probably be useful to cite Fujiwara et al. (2017), the overview paper for the SPARC Reanalysis Intercomparison Project, here -- they provide a detailed description of all the commonly used reanalyses.

While not critical to this paper, it would be interesting to see how the bifurcation changes and where it is present at levels above 850K.  I realize that ERA-Interim isentropic level PV data are not provided above 850K, but a few isentropic levels in the upper stratosphere are provided in the CFSR dataset.  Could you use that to make a brief comment as to what we might expect at higher altitudes?

*Section 2.2:*

Page 3, lines 9--13:  The description of this procedure is not clear to me.  As I read this, you are using the PV data as distributions from which to find, at each timestep and vertical level, what PV value each of 181 linearly spaced percentile values corresponds to, and then calculating the EL of each of the resulting 181 PV values?  Since PV as a function of EL is calculated (by whatever method) before the gradients with respect to EL are taken, how could this lead to spurious zero values of PV gradient?  Since the fact that there are regions (which vary from day to day) with much weaker and stronger PV gradients means that the resultant PV values at which you find EL will not be uniformly spaced, it appears to me that using the percentile steps might lead to a very non-uniform coordinate mapping (from lat/lon to EL) and one that changes from day to day and level to level -- could this then have a significant effect on the subsequent numerical derivatives to get the PV gradients?  I apologize if I am off the mark with these questions -- if I am, I suspect that also argues for clarifying the description.

Page 3, line 16:  Is 40deg EL always low enough for the SH polar vortex?  This may not be a problem at the middle and lower stratospheric levels you are looking at here, but from the vortex edge diagnostics I've looked at, in the upper stratosphere, especially in the SH, the vortex is often very large, and I frequently found peaks in PV gradients or windspeed x PV gradient that were slightly equatorward of 40deg.

Page 3, lines 16--19:  Is anything done to eliminate spurious peaks in this primary and secondary peak identification?  One of the advantages of using windspeed x PV gradients or constraining PV gradient maxima to be near windspeed maxima is reducing the occurrence of such peaks.  Can you say anything about how much of an issue this might be, especially, eg, where you find widely separated peaks and one is near 80 EL where spurious peaks are more common?  Also, are any other criteria used (e.g., a minimum windspeed, such as in Nash et al, 1996 and Manney et al, 2007; and/or a minimum PV gradient such as in Manney et al, 2007) to determine when the vortex is well-defined enough to analyze (you do mention a minimum PV gradient criterion in a figure caption later; if this is applied in general, it might be good to move it to this methods description and explain the reasoning behind the choice of value)?

Page 3, lines 20 to 28 and Table 1: The rationale behind the choices of combinations of peak ratio and dip fraction for the bifurcation index isn't clear, could you please explain this further? Why does it make sense for the dip fraction to go up as the peak ratio goes up? What did you look at to determine these values (e.g., extensive visual inspection and of what if so, or some other measures)? When you say on lines 24--25 that the dip ratio has more influence on BI, why is that desireable (presumably it is or you wouldn't have made the choices you did)?

How are the various bifurcation diagnostics (BI, as well as peak ratio and dip fraction) affected by changes in EL resolution (e.g., using 91 or 361 percentile steps)? Would such a change significantly affect your results (including the thresholds used to determine BI)?

Figure 1: You might consider using thicker lines and/or bolder colors; while it is OK on a good high-resolution monitor, on many printers or lower resolution monitors, the red line in the lower panel will be barely visible. (A similar comment applies to Figure 9.)

*Section 3.1:*

Page 5, line 9: What is meant by "the inside and outside edges of the region of maximum wind speed"? Since the maximum wind speed is a point (on each isentropic level), the "region" of maximum wind speed is not defined unless you define it, which must be done before you can define the inside and outside of it and whether something is near that. (This occurs again later in the paper as well.)

Figure 2: Please specify in the caption that these are monthly means (I believe that's true).

*Section 3.2:*

Figure 4: It might be helpful to put horizontal lines at the boundaries of what you call "mid-levels", since those are mentioned several times in the text. Also, what is the reasoning behind the choice of the PV gradient threshold?

*Section 3.3:*

Page 7, line 5 and Figure 5: It isn't at all clear to me that Figure 5 shows the peaks to be "more dispersed in August (which I would take to be "winter" rather than "Spring") than September, if that is what you mean by this statement. If by "earlier in the season" you mean before August, why not show July as well to demonstrate this?

*Section 3.4:*

Page 8, line 2: Can you be more specific than "in some years"? How many of the years do you see it in during a significant part of the month?

Figure 6 shows a sharp change between 430 and 475K in September, which can also be seen in Figure 5 in August and September.  Do you have any thoughts on why this might be?

It would be nice to see August and October in Figure 6 as well as the three months shown, especially since the months that are shown look very different -- what does the transition between these regimes look like?  Or at least describe this in the text.

In Figure 6, it is very hard to distinguish the lowest 2-3 values from the zero values that represent times without a sufficiently well-developed vortex -- perhaps you could adjust the color palette and/or find another way to clearly indicate "missing data".

*Section 4.1:*

It would be helpful to have the CFSR plots together with the ERA-Interim ones, to facilitate comparison; this comparison also seems like something that it would be good to show earlier in the paper.

*Section 4.2:*

Figure 9 and discussion:  It would be very helpful to show the gradients of relative vorticity, absolute vorticity, and d(theta)/dp -- it is difficult to judge a change in the gradients from the fields themselves.  Also, it would be much more convincing (perhaps the same term is not the dominant factor in all cases? Or not as dominant for all BI values?) if this analysis was done for some sort of climatology or composite of cases with bifurcation rather than a single day in a year that may highlight a particular regime of bifurcation.

Page 11, line 3, I'd suggest saying something like "...provide context for recent *reports of* trends in ozone…", since the results of Solomon et al (2016) are controversial.

*Section 4.3:*

Page 11, lines 6--9: Not necessarily something to change in the paper, but I just note that such large variations from day to day are fairly common even in cases where there is no bifurcation (e.g., in the NH -- you can see examples of this in pretty much any winter season on some isentropic levels, eg, in the vortex edge PV and area routinely posted on https://ozonewatch.gsfc.nasa.gov/meteorology/NH.html
-- these are calculated using the method of Nash et al, 1996, but other PV gradient-based methods can suffer from the same problem).  This is one reason why a constant climatological value is sometimes preferred over an automated day-to-day calculation for applications where a single value for the vortex edge is needed (e.g., Manney et al., 2007; Manney and Lawrence, 2016; Lawrence and Manney, 2017).

Page 12, lines 6--7 and Figure 10: It would be interesting (and possibly a better demonstration) to compare this with what the inner and outer edges of the vortex boundary region look like when the method of Nash et al. (1996) is used without taking the bifurcation into account.

**Typographical errors, minor wording suggestions:**

Page 1, line 9: suggest changing "reduces" to "decreases" ("reduces" is not commonly used as a verb).

Page 3, line 9: "across the polar vortex" sounds a bit odd, since you are calculating the PV gradients over an EL range that includes both vortex and extra-vortex regions. Perhaps reword this.

Page 3, line 23: When you say "profiles" (this also occurs elsewhere in the paper), do you mean the PV gradient vs EL curves for each time? This may seem a trivial point, but can be confusing to those of us who automatically think "vertical" when they see the word "profile"! Perhaps you could orient those of us who are thinking in a different direction by, the first time you use the terminology, saying something like "profiles of PV gradient as a function of EL".

Page 6, line 9, "display" should be "displays" (or "wind speed" should be "wind speeds").

Page 6, line 12, suggest "decreases" instead of "reduces"

Figure 6 caption, should be "Data are", not "Data is"

Page 9, line 4, "pause" (implies a time variable) doesn't seem like the right word here.

**References not already cited in this manuscript:**

Ajtić, J.; Connor, B. J.; Lawrence, B. N.; Bodeker, G. E.; Hoppel, K. W.; Rosenfield, J. E. & Heuff, D. N. Dilution of the Antarctic ozone hole into southern midlatitudes, 1998-2000, *J. Geophys. Res.*, 109, 2004.

Fujiwara, M., J.S. Wright, G.L. Manney, L.J. Gray, J. Anstey, T. Birner, S. Davis, E.P. Gerber, V.L. Harvey, M.I. Hegglin, C.R. Homeyer, J.A. Knox, K. Kr ¨uger, A. Lambert, C.S. Long, P. Martineau, B.M. Monge-Sanz, M.L. Santee, S. Tegtmeier, S. Chabrillat, D.G.H. Tan, D.R. Jackson, S. Polavarapu, G.P. Compo, R. Dragani, W. Ebisuzaki, Y. Harada, C. Kobayashi, W. McCarty, K. Onogi, S. Pawson, A. Simmons, K. Wargan, J.S. Whitaker, and C.-Z. Zou, Introduction to the SPARC Reanalysis Intercomparison Project (S-RIP) and overview of the reanalysis systems, *Atmos. Chem. Phys.*, doi:10.5194/acp-17-1417-2017, 1417–1452, 2017.

Lary, D. J.; Chipperfield, M. P.; Pyle, J. A.; Norton, W. A. & Riishojgaard, L. P., Three-dimensional tracer initialization and general diagnostics using equivalent PV latitude-potential-temperature coordinates, *Q. J. R. Meteorol. Soc.*, 121, 187-210, 1995.

Lawrence, Z.D., and G.L. Manney, Characterizing stratospheric polar vortex variability with computer vision techniques, *J. Geophys. Res.*, in press, 2017.

Manney, G.L., Z.D. Lawrence, M.L. Santee, W.G. Read, N.J. Livesey, A. Lambert, L. Froidevaux, H.C. Pumphrey, and M.J. Schwartz, A minor sudden stratospheric warming with a major impact: Transport and polar processing in the 2014/2015 Arctic winter, *Geophys. Res. Lett.,* 42, 7808–7816, doi:10.1002/2015GL065864, 2015.

Manney, G. L., Z. D. Lawrence, M. L. Santee, N. J. Livesey, A. Lambert, and M. C. Pitts, Polar processing in a split vortex: Arctic ozone loss in early winter 2012/2013, *Atmos. Chem. Phys.*, 15(10), 5381–5403, doi:10.5194/acp-15-5381-2015, 2015.

Manney, G.L., et al., Solar Occultation Satellite Data and Derived Meteorological Products: Sampling Issues and Comparisons with Aura MLS, *J. Geophys. Res.* 112, D24S50, 10.1029/2007JD008709, 2007.

Manney, G.L., M.L. Santee, N.J. Livesey, L. Froidevaux, H.C. Pumphrey, W.G. Read, and J.W. Waters, EOS Microwave Limb Sounder Observations of the Antarctic Polar Vortex Breakup in 2004, *Geophys. Res. Lett.*, 32, L12811, doi:10.1029/2005GL022823, 2005.

Manney, G.L., J.L. Sabutis, D.R. Allen, W.A. Lahoz, A.A. Scaife, S. Pawson, C.E. Randall, B. Naujokat, and R. Swinbank, Simulations of dynamics and transport during the September 2002 Antarctic major warming, *J. Atmos. Sci.*, 62, 690–707, 2005.

Manney, G.L., H.A. Michelsen, M.L. Santee, M.R. Gunson, F.W. Irion, A.E. Roche, and N.J. Livesey, Polar vortex dynamics during spring and fall diagnosed using trace gas observations from the Atmospheric Trace Molecule Spectroscopy instrument, *J. Geophys. Res*., 104, 18,841-18,866, 1999.

Rummukainen, M., B. Knudsen, and P. von der Gathen, Dynamical diagnostics of the edges of the polar vortices, *Annales Geophysicae*, 12, 1114–1118, 1994.

Schoeberl, M. R., M. Luo, and J. E. Rosenfield, An analysis of the Antarctic Halogen Occultation Experiment trace gas observations, *J. Geophys. Res.,* 100(D3), 5159–5172, doi:10.1029/94JD02749, 1995.

Schoeberl, M. R., L. R. Lait, P. A. Newman, and J. E. Rosenfield, The structure of the polar vortex, *J. Geophys. Res.*, 97, 7859–7882, 1992.

Smith, M. & McDonald, A. J., A quantitative measure of polar vortex strength using the function *M*, J. Geophys. Res., 119, 5966-5985, 2014.

Trounday, B., L. Perthuis, S. Strebelle, J. D. Farrara, and C. R. Mechoso, Dispersion properties of the flow in the southern stratosphere during winter and spring, *J. Geophys. Res.*, 100(D7), 13,901–13,917, doi:10.1029/95JD00774, 1995.

---

## Short Comment (SC2) · 1 Dec 2017

Dear authors,

I think that it is absolutely vital for your study to reconsider your results with respect to the recently published work by Serra et al. (2017). Their geodesic method identifies a materially optimal vortex boundary. More to that, Serra et al. (2017) show also a comparison of their method with a "Nash" vortex boundary. To cite some of their conclusions: "Nash method is frame dependent and nonmaterial, hence a priori unsuitable for a self-consistent detection of coherent transport barriers." "Given its Eulerian (non-material) nature, the Nash edge evolves discontinuously, with visible jumps in position

and shape over time."

My personal opinion would be that your analysis of PV gradients may be untouched, but the relationship to mixing barriers needs to tackle these new findings.

Best regards, Petr Šácha

sacha@uvigo.es

Uncovering the Edge of the Polar Vortex Serra, M., P. Sathe, F. Beron-Vera, and G. Haller Journal of the Atmospheric Sciences 2017 74:11, 3871-3885

---

## Referee Comment (RC2) · G. Manney (Referee) · 15 Dec 2017

The method described by Serra et al (2017) is an interesting addition to the suite of methods used to determine the approximate location of the vortex edge, a task that is commonly useful and/or necessary. It will be interesting in the future to see this method validated, since Serra et al show only one 12-day period in the NH winter during a dynamical regime when any reasonable definition of the vortex edge would give satisfactory results for the vast majority of studies, and since they did not present any quantitative evidence that it corresponds better with the location of the transport barrier than other methods. It will also be interesting to see it applied to the vortex in the SH,

where (as Conway et al, as well as previous work, have clearly demonstrated) the vortex and its "edge" have a very different character than in the NH. It may be appropriate for Conway et al to cite Serra et al, for example, in the introduction where they mention numerous methods of defining the vortex edge, including other Lagrangian methods, and/or in the discussion where they mention the desirability of exploring how the bifurcation of the PV gradients in the vortex edge region relates to mixing diagnostics and other Lagrangian methods. However:

(1) The focus of Conway et al, as I read it – and in my opinion the thing that makes their work valuable and unique – is not on determining a single vortex edge by any method, but on describing and understanding the structure of the flow in the whole "vortex edge region", of which PV and its gradients provide a physically-based description. Any work, such as that of Serra et al, that focuses purely on defining a single vortex edge, is really peripheral to the main points of Conway et al, so I don't see it as essential to add any further discussion of relationships to any of the numerous possible vortex edge definitions that have been developed, even those that have already been shown to be useful in practical studies and/or have been validated in relation to the transport barriers; in fact such discussion might distract from the main points of the paper.

(2) Many, if not most, studies where a single vortex edge definition / contour is needed require that value to be computed for many years and many levels in both hemispheres – studies of, e.g., approximately 40 years of reanalysis data or hundreds of years of climate model data, on many levels throughout the stratosphere – they must thus use a method that allows the vortex edge to be calculated quickly and efficiently for such vast volumes of data. Therefore, methods such as that of Serra et al, and other methods based on, e.g., Lagrangian descriptors or other computationally-intensive mixing diagnostics, are not expected to be the ones widely used in most practical applications, although they may prove to be very useful in evaluating these more simple and practical methods. The exploration by Conway et al. of how one of those practical/efficient methods might be modified to describe the bifurcated SH vortex edge more accurately

is therefore much more to the point of developing a practical diagnostic that more accurately describes the vortex edge region in the SH.

---

## Author Comment (AC1) · 27 Apr 2018

*Note: reviewer comment and short comment text is given in blue, while the author response is given in black type underneath.*

**Response to reviewer comment by Gloria L Manney on "Bifurcation of potential vorticity gradients across the Southern Hemisphere stratospheric polar vortex."**

**Jono Conway**

We thank Prof. Manney for her helpful and constructive comments. We have made many minor additions and changes to the text to clarify the background and methods. Her comments also led us to extend the method to detect bifurcation to more explicitly exclude periods when the vortex is not well developed enough to analyse. The revised figures 4-12 now further highlight the strong bifurcation in the mid-stratosphere during late winter and spring. We have responded to each of the general and specific comments below.

General Comments:

This paper uses ERA-Interim reanalysis data to describe the climatological bifurcation of potential vorticity gradients in the Antarctic vortex edge region. Overall, the analysis is sound and the paper is well-written. To my knowledge, a detailed description/climatology of the bifurcation of PV gradients in the Antarctic vortex edge region has not been published, and the material will thus be of interest to ACP readers. In its current form, the paper is somewhat lacking in referencing previous work related to vortex edge definitions and mixing barriers, and in explaining the rationale behind the definitions of bifurcation index and other threshold choices used. The introduction, in particular, doesn't provide a clear picture of the relevant background and previous work. Once these issues, and some other specific (minor) points listed below, have been addressed, the paper will be appropriate for publication in ACP.

We have added further background material to the introduction and clarify the choices behind various aspects of the methodology in the text – see specific comments below.

Note 1: Obviously, Zachary Lawrence and I work together. However, we were each notified about the paper separately and each read the paper and wrote up our comments that are posted here independently.

Note 2: I am not suggesting that you cite every one of the numerous Manney et al. papers mentioned in this review; it is inevitable that I am most familiar with those I wrote, and in some cases they may lead you to more appropriate citations; in a couple of cases they are just mentioned as "of interest".

Specific Comments (in order of appearance in the paper):

Introduction:

Overall, the authors should make clear that much of this discussion (and which parts of it) applies to the Antarctic polar vortex only; many of the statements made are either not applicable to the Arctic vortex, or would need to be qualified (e.g., the discussion of ozone loss and dispersal from the vortex)

to describe the Arctic; this is fine -- the paper focuses on the Antarctic -- but should be said explicitly when there are statements that apply only to the Antarctic. In addition, some statements are rather vague, may not be quite accurate, and/or could use more appropriate citations. Specifically:

Page 1, lines 20--21: "...cold air formed in the polar night…" -- "formed" doesn't seem quite accurate. Changed to "In the Southern Hemisphere, during winter and spring, the barrier isolates the cold stratospheric air that develops during the polar night from warmer mid-latitude air."

● Page 1, line 19: Schoeberl et al (1992) would be a good reference to add to those here; also please use "e.g.," ahead of the reference list in cases like this where there are many papers relevant to the statement. "e.g." added to reference list here and elsewhere. Reference to Schoeberl et al (1992) added.

● Page 1, line 22: Add "e.g.," before or "and references therein" after "(Solomon, 1999)"

'and references therein' added

● Page 1, lines 23--24: Please emphasize that this applies to the Antarctic vortex. "Southern Hemisphere" added before 'polar vortex' on line 23.

● Page 2, lines 1--2: There are many, some earlier, references that could be used here for both statements (I'm sure that is also the case for the statement on line 3, though I am not well-read on such CCM studies). At the very least, "e.g.," or "and references therein" should be used. Some other useful papers to cite (including only ones specifically relevant to the SH) might be:

○ Schoeberl et al, 1995 (JGR)

○ Lary et al, 1996 (QJ)

○ Manney et al, 1999 (JGR), 2005 (JAS), 2005 (GRL)

○ Ajtić et al, 2004 (JGR)

Additional references added.

● Page 2, lines 4--18: There are some vortex edge definition methods that are not mentioned here, as well as relevant papers It would be good to cite to briefly discuss other ways the vortex edge or boundary region can be defined besides those you already mention.

The discussion of methods to define the vortex edge(s) has been expanded with many new references.

Also, the statement that "Bodeker et al. (2002) formalised this method" is not accurate: Determining the vortex boundary using the maximum in windspeed x PV gradient is not the same as determining it by the maximum PV gradients but requiring that maximum to be in proximity to the windspeed maximum (see, e.g., the brief discussion in Manney et al., 2007, who implemented and compared the two methods, starting with code provided by Eric Nash for the method described in Nash et al).

The sentence has been changed to:

"The centre of the vortex boundary region is most often defined as …. the EL of the maximum of the PV gradient times the average wind speed (e.g. Bodeker et al 2002, Manney et al 2007)"

It would also be good to mention some other papers that discuss different ways of defining the vortex edge, especially some of the early papers that use PV gradients (eg, Trounday et al, 1995) and/or use other methods and compare with PV gradient methods (eg, Paparella et al, 1997, JGR, already cited

here but the comparisons are not mentioned; Rummukainen et al, 1994, Ann. Geophys.) Smith and McDonald (2014, JGR) used the "function M " to look at the strength of the Antarctic vortex and the width (area) of the vortex boundary region. The authors may be interested (though certainly not obliged to cite a paper that will only be available online sometime in the next few days after I'm writing this review) in the vortex edge method in Lawrence and Manney (2017, JGR, in press), which also gives further discussion of various vortex edge definition methods and literature.

These and other references have been added.

● Page 2, line 12: Add "e.g.," before "Manney and Lawrence, 2016" (I am sure others have done this as well; our two Manney et al (2015) papers are ones where I know this was done). changed to e.g. Waugh and Randel, 1999; Manney and Lawrence, 2016

● Page 2, lines 11--18: May be worth noting that Manney and Lawrence (2016) compared their climatological selection of vortex edge PV values to daily PV gradients, effective diffusivity, the function M, and trace gas gradients. This section has been reworked, so a comment is now not required.

Section 2.1:

It would probably be useful to cite Fujiwara et al. (2017), the overview paper for the SPARC Reanalysis Intercomparison Project, here -- they provide a detailed description of all the commonly used reanalyses. Added in last paragraph of introduction.

While not critical to this paper, it would be interesting to see how the bifurcation changes and where it is present at levels above 850K. I realize that ERA-Interim isentropic level PV data are not provided above 850K, but a few isentropic levels in the upper stratosphere are provided in the CFSR dataset. Could you use that to make a brief comment as to what we might expect at higher altitudes?

We prefer to maintain ERA-interim as the primary dataset, so it does not make sense to present these data here. As noted in other papers (e.g. Manney and Lawerence, 2016), PV gradients are less well defined in the upper stratosphere, so I expect that the method to detect bifurcation may be less valid at these altitudes. The introduction has been amended to emphasize the focus on the mid-stratosphere where the highest rates of ozone depletion occur.

Section 2.2:

Page 3, lines 9--13: The description of this procedure is not clear to me. As I read this, you are using the PV data as distributions from which to find, at each timestep and vertical level, what PV value each of 181 linearly spaced percentile values corresponds to, and then calculating the EL of each of the resulting 181 PV values? Yes – see full explanation below. Since PV as a function of EL is calculated (by whatever method) before the gradients with respect to EL are taken, how could this lead to spurious zero values of PV gradient?

Spurious zero values of PV gradient can occur when equally-spaced PV steps are chosen to calculate EL. Where no PV data falls between the two PV steps, both PV steps are assigned the same EL. In these cases, a spurious zero PV gradient occurs. This is especially the case with CFSR where the data is truncated to a reduced precision. At some levels, the PV data in CFSR has only a few hundred unique values and this creates a significant problem for EL analysis.

Since the fact that there are regions (which vary from day to day) with much weaker and stronger PV gradients means that the resultant PV values at which you find EL will not be uniformly spaced, it appears to me that using the percentile steps might lead to a very non-uniform coordinate mapping

(from lat/lon to EL) and one that changes from day to day and level to level -- could this then have a significant effect on the subsequent numerical derivatives to get the PV gradients? I apologize if I am off the mark with these questions -- if I am, I suspect that also argues for clarifying the description.

In addition to the spurious zero values discussed above, using equally-spaced PV also produces ringing in the CFSR dataset, as the data is truncated to a reduced precision during the analysis using a multiplier that is not always appropriate. To avoid using equally-spaced PV steps, our two options are thus: map the EL of every unique PV value, or find a way to interrogate a useful and reliable subset of PV values. Mapping the EL of every unique PV value is computationally expensive and increases the chances of ringing in the derivatives (as discussed by Nash et al 1996). To be useful, the data can be resampled to regular EL, but this step may introduce its own issues.

We have found the most reliable way to choose a subset of PV values is through percentile steps. Here, we chose to sample 181 percentile values, as this is similar to the number of latitude points per hemisphere in the reanalysis datasets. Each hemisphere is treated separately and for the SH, PV values are multiplied by -1 to ensure a common treatment with NH. To capture the extreme tail of the PV distribution, 8 steps are placed between 99th percentile and maximum value [99, 99.5, 99.75, 99.875, 99.9375, 99.96875, 99.984375, 100th percentiles]. The remaining 173 bins are spread linearly between the minimum (i.e. the most positive PV values in the SH) and 99th percentile. In essence, our method simply takes an a-priori guess at a regular EL spacing (by assuming all grids points have the same area), then chooses PV steps based on that. Other methods assume constant PV spacing to arrive at the same gradients.

We have thought about whether using percentile steps will have an adverse effect on the derived PV gradients, but have not found any evidence that it degrades the quality of the analysis. To illustrate the differences between our method and others, the PV-EL relationship was calculated using percentile steps and equally-spaced steps (denoted linear in Figure 1) for an example time/height (Figure 1; 1 September 2007 at 550K in CFSR). The PV-EL relationship was also calculated at constant steps of 0.1 sPVU or by using all unique PV values, which are then resampled at 0.5 degree EL steps. Figure 1 shows the PV, 1st and 2nd derivatives of PV with respect to EL, PV and EL spacing, and average wind speed for these different methods.

Like any other method, using percentile steps to interrogate the PV-EL relationship will result a different coordinate mapping from day to day and level to level. While the PV spacing when using the percentile method is non-uniform, the EL spacing is close to uniform. Aside from small differences in the precise EL of each step, all methods return very similar PV, wind speed. The gradients returned using percentile steps or all unique PV values show only minor differences and this gives us confidence the results are not sensitive the choice of a particular method. The percentile method has the advantages that PV gradients can be derived directly from the steps used to create the PV-EL mapping, rather requiring the PV-EL relationship to be resampled to reasonable resolution (0.5 degree). Equally-spaced steps show ringing in the first derivative that becomes much worse in the 2nd derivate.

We have updated the description of the method in the manuscript to include more details about the percentile method, along with a comment about the effect on the derivatives obtained.

[Figure]

**Figure 1. Equivalent latitude calculations using various methods to choose PV steps. All have 181 steps. Percentile = method used in paper; linear = 181 steps evenly-spaced between minimum and maximum PV, 0.1PVU = EL calculated for 0.1 sPVU steps, then distribution resampled to 0.5degree EL steps; all = the same as 0.1PVU but calculated for every unique PV value. This example is for 1 September 2007 at 550K in CFSR.**

Page 3, line 16: Is 40deg EL always low enough for the SH polar vortex? This may not be a problem at the middle and lower stratospheric levels you are looking at here, but from the vortex edge diagnostics I've looked at, in the upper stratosphere, especially in the SH, the vortex is often very large, and I frequently found peaks in PV gradients or windspeed x PV gradient that were slightly equatorward of 40deg.

It is likely some peaks are missed at higher levels during mid-winter, but for simplicity a single latitude range was chosen here. An updated Figure 5 shows that peaks close to 40° S are not common. Only at 850K during May through August is it likely for peaks to be missed due to the 40° cut-off; including both primary and secondary peaks, 1.7%, 5.5%, 6.8% and 1.9% of peaks at 850K in these months are

found north of 42.5 degrees, respectively.  At other levels much less than 1% of peaks are found close to the 40° cut-off.

 Is anything done to eliminate spurious peaks in this primary and secondary peak identification? One of the advantages of using windspeed x PV gradients or constraining PV gradient maxima to be near windspeed maxima is reducing the occurrence of such peaks. Can you say anything about how much of an issue this might be, especially, eg, where you find widely separated peaks and one is near 80 EL where spurious peaks are more common?  Also, are any other criteria used (e.g., a minimum windspeed, such as in Nash et al, 1996 and Manney et al, 2007; and/or a minimum PV gradient such as in Manney et al, 2007) to determine when the vortex is well-defined enough to analyze (you do mention a minimum PV gradient criterion in a figure caption later; if this is applied in general, it might be good to move it to this methods description and explain the reasoning behind the choice of value)?

We now introduce a minimum wind speed of 15.2 m/s (Nash et al. 1996) and minimum PV gradient (significantly above average; see methods for calculation) to constrain the detection of primary and secondary peaks, and to determine if the vortex is well developed enough to analyse. Less than 2% of all peaks are found south of 77.5° S at all levels, so spurious peaks near to 80°S are likely not common. The methods section has been updated to reflect this.

Page 3, lines 20 to 28 and Table 1: The rationale behind the choices of combinations of peak ratio and dip fraction for the bifurcation index isn't clear, could you please explain this further? Why does it make sense for the dip fraction to go up as the peak ratio goes up? What did you look at to determine these values (e.g., extensive visual inspection and of what if so, or some other measures)? When you say on lines 24--25 that the dip ratio has more influence on BI, why is that desireable (presumably it is or you wouldn't have made the choices you did)?

The threshold values used for BI were chosen to capture cases with clear bifurcation after extensive visual inspect on PV gradients. For this reason, only peaks with at least 50% the PV gradient of the main peak are considered bifurcated. In the same way, if there is no clear dip between the two peaks, a second maximum is more likely a 'shoulder' on the main peak or noise in the PV gradient. The 5% minimum on the dip fraction was considered large enough to avoid these situations.

The two thresholds increase at the same time so that strong bifurcation requires both a similarly sized secondary peak and a very clear dip between. Without a strong dip, a second peak of similar magnitude more likely represents a wide area of strong PV gradient (or a shoulder on the main peak) rather than a distinct peak in its own right. For this reason, it is desirable for the dip ratio to have a stronger influence on BI – it weeds out the more indistinct peaks.

The threshold values were determined by varying linearly between our chosen extremes for peak ratio (50% and 95%), and the corresponding minimum and maximum peak ratio (5% and 50%). Other combinations of threshold values may also give satisfactory results, but, as the metric is only intended to be a guide, this is not a concern. The BI metric is complemented by the peak ratio and dip fraction, which allow a more nuanced description of a bifurcated profile if desired. The primary reason for the BI metric is to detect the presence of bifurcation and to allow inter-annual variations in bifurcation to be analysed.

Further explanation of these choices is now given in the methods.

How are the various bifurcation diagnostics (BI, as well as peak ratio and dip fraction) affected by changes in EL resolution (e.g., using 91 or 361 percentile steps)? Would such a change significantly affect your results (including the thresholds used to determine BI)?

Changing the resolution to 90 or 360 steps results in very similar patterns of BI, dip fraction and peak ratio, but the frequency of bifurcation is increased (Figure 2). The magnitude of the average peak ratio is not substantially changed with changing resolution, whereas dip fraction increases with increasing resolution. At higher resolutions, the increased dip fraction and more frequency bifurcation increases average BI, but the general pattern is the same. If using a different resolution, the dip-ratio threshold used to define BI may need to be changed, but the story will remain the same. The PV gradient thresholds (mean + 2SD) at 90 and 360 steps are within 1% of the values at 181 steps, so are not sensitive to resolution.

[Figure]

**Figure 2. Monthly average BI, fraction of profiles bifurcated, dip fraction and peak ratio (columns from left to right) for 90 (top row) , 181 (middle row) , and 360 (bottom row) percentile steps at selected levels in ERA-Interim [compare to Figure 4 a,e,f].**

 Line weight in Figures 1 and 9 has been increased.

Section 3.1:

Page 5, line 9: What is meant by "the inside and outside edges of the region of maximum wind speed"? Since the maximum wind speed is a point (on each isentropic level), the "region" of maximum wind speed is not defined unless you define it, which must be done before you can define the inside and outside of it and whether something is near that. (This occurs again later in the paper as well.)

We have clarified this sentence "While increased climatological PV gradients occur over a broad region, at lower levels these are nearer the poleward edge of the wind speed jet, while at higher levels these are nearer the equatorward edge of the jet".

Figure 2: Please specify in the caption that these are monthly means (I believe that's true).

Yes, these are monthly means of PV gradients in 0.5 degree bins. Caption amended.

Section 3.2:

Figure 4: It might be helpful to put horizontal lines at the boundaries of what you call "mid-levels", since those are mentioned several times in the text.

We prefer not to add horizontal lines as these would unduly influence how the reader interprets the figure. i.e. we can have suggested a priori what the pattern they should see is.

Also, what is the reasoning behind the choice of the PV gradient threshold?

We previously used a value of 0.45 sPVU to filter the climatology as it was 1.5 times the PV gradient that we had assessed (in other work) as being associated with the boundaries of the vortex edge region. We now introduce a PV gradient threshold into the calculation of the BI metrics and do not use this criterion to filter the climatology. The PV gradient criterion is now defined as the mean + 2*standard deviation of the PV gradient between 5 and 80 degrees at a given level, assessed using profiles from the entire timeseries in all months. We now only filter the monthly average climatology (Figure 4) for periods when bifurcation is present in less than 1% of profiles. We filter the timeseries of monthly averages (Figure 6) when the vortex is not well developed in less than 50% of time periods. The methods section and figure captions have been updated to reflect this.

Section 3.3:

Page 7, line 5 and Figure 5: It isn't at all clear to me that Figure 5 shows the peaks to be "more dispersed in August (which I would take to be "winter" rather than "Spring") than September, if that is what you mean by this statement. If by "earlier in the season" you mean before August, why not show July as well to demonstrate this?

We have clarified this statement to "At 700K and 850K, the locations of primary peaks are spread across a wider range of latitude bins, especially in August."

Section 3.4:

Page 8, line 2: Can you be more specific than "in some years"? How many of the years do you see it in during a significant part of the month?

Average BI > 0.75 in November is only seen in two years (1987 at 475K and 530K, and 2015 at 600K).

Figure 6 shows a sharp change between 430 and 475K in September, which can also be seen in Figure 5 in August and September. Do you have any thoughts on why this might be?

At 430K, secondary peaks are not common (Figure 4b) and are usually closer to the primary peak (Figure 4c) and weaker than at 475K (Figure 4e). There is a shift in the preferred location of secondary peaks – equatorward at 475 and poleward to 430. The mechanism for this change is likely related to the fact that the PV gradients on the inside edge of the vortex boundary at 430K don't strengthen during September as they do between 475K and 600K (See figure 2 and 3). I suspect that the cause of the enhanced PV gradient from 475 to 600K is related to changes in ozone concentrations and temperature, but investigating this would be beyond the scope of this paper.

It would be nice to see August and October in Figure 6 as well as the three months shown, especially since the months that are shown look very different -- what does the transition between these regimes look like? Or at least describe this in the text.

We have added the months August and October to Figure 6 and updated the text to describe the transition between these months. To save space we have also removed July and November, as they are largely redundant (November shows almost no values of BI > 1 and August is very similar to July). The months shown in Figure 6 now align with Figures 5 and 10.

In Figure 6, it is very hard to distinguish the lowest 2-3 values from the zero values that represent times without a sufficiently well-developed vortex -- perhaps you could adjust the color palette and/or find another way to clearly indicate "missing data".

We have introduced a grey colour to denote periods when the vortex is not well developed (more than 50% of profiles within a month do not meet the minimum ws and PV gradient thresholds). We have updated the captions.

Section 4.1:

It would be helpful to have the CFSR plots together with the ERA-Interim ones, to facilitate comparison; this comparison also seems like something that it would be good to show earlier in the paper.

A full formal comparison of ERA-Interim and CFSR is not possible, as each data set has different isentropic levels available for download. We prefer to simply use CFSR to confirm the patterns observed in the ERA-Interim data set are real phenomena, rather than focus on comparing all of the results using two reanalysis data sets. For these reasons, we would prefer to keep the section separate. This same format is also used in recent papers on stratospheric mixing (e.g. Abalos et al, 2016). We clarify our choice to focus on results from the higher-vertical-resolution ERA-Interim data set in the methods section.

Section 4.2:

Figure 9 and discussion: It would be very helpful to show the gradients of relative vorticity, absolute vorticity, and d(theta)/dp -- it is difficult to judge a change in the gradients from the fields themselves. Also, it would be much more convincing (perhaps the same term is not the dominant factor in all cases? Or not as dominant for all BI values?) if this analysis was done for some sort of climatology or composite of cases with bifurcation rather than a single day in a year that may highlight a particular regime of bifurcation.

We have updated Figure 9 with the gradients of vorticity and static stability with respect to EL, so that the gradients can be directly observed. We have also provided a new figure (new Figure 10) showing height-latitude composites of vorticity and static stability gradients, averaged from the 5 years with the highest BI at mid-levels. The composite analysis reinforces the findings from the case study day, showing BI is associated with elevated vorticity gradients in two locations, along with increased static stability gradients at the inner location.

Page 11, line 3, I'd suggest saying something like "...provide context for recent *reports of* trends in ozone…", since the results of Solomon et al (2016) are controversial. wording changed to 'for recent reports of trends'

Section 4.3:

Page 11, lines 6--9: Not necessarily something to change in the paper, but I just note that such large variations from day to day are fairly common even in cases where there is no bifurcation (e.g., in the NH -- you can see examples of this in pretty much any winter season on some isentropic levels, eg, in the vortex edge PV and area routinely posted on https://ozonewatch.gsfc.nasa.gov/meteorology/NH.html -- these are calculated using the method of Nash et al, 1996, but other PV gradient-based methods can suffer from the same problem). This is one reason why a constant climatological value is sometimes preferred over an automated day-to-day calculation for applications where a single value for the vortex edge is needed (e.g., Manney et al., 2007; Manney and Lawrence, 2016; Lawrence and Manney, 2017).

These large day to day variations that we were observing in our analysis of the vortex edge were part of the motivation for this study, and the method presented here partly overcomes this issue. A new figure (new Figure 11) is introduced to demonstrate the difference in the temporal evolution of the vortex edges with and without taking into account bifurcation.

Page 12, lines 6--7 and Figure 10: It would be interesting (and possibly a better demonstration) to compare this with what the inner and outer edges of the vortex boundary region look like when the method of Nash et al. (1996) is used without taking the bifurcation into account.

We have added extra panels to Figure 10 (now Figure 12) to show the position of the inner and outer edges without taking into account bifurcation. Along with the new Figure 11 showing the temporal evolution of the edges in 2007, the added benefit of the BI algorithm is now more clearly shown.

Typographical errors, minor wording suggestions:

Page 1, line 9: suggest changing "reduces" to "decreases" ("reduces" is not commonly used as a verb). Changed to "decreases"

Page 3, line 9: "across the polar vortex" sounds a bit odd, since you are calculating the PV gradients over an EL range that includes both vortex and extra-vortex regions. Perhaps reword this. Changed to "Profiles of PV gradient as a function of EL were calculated…"

Page 3, line 23: When you say "profiles" (this also occurs elsewhere in the paper), do you mean the PV gradient vs EL curves for each time? This may seem a trivial point, but can be confusing to those of us who automatically think "vertical" when they see the word "profile"! Perhaps you could orient those of us who are thinking in a different direction by, the first time you use the terminology, saying something like "profiles of PV gradient as a function of EL". References to this terminology has been added to the first two paragraphs of the section 2.2

Page 6, line 9, "display" should be "displays" (or "wind speed" should be "wind speeds"). Changed to 'display'

Page 6, line 12, suggest "decreases" instead of "reduces" Changed to 'decreases'

Figure 6 caption, should be "Data are", not "Data is" Sentence has been removed from caption

Page 9, line 4, "pause" (implies a time variable) doesn't seem like the right word here. "Pause" changed to "weakening"

**Response to short comment by Zachary D. Lawrence on "Bifurcation of potential vorticity gradients across the Southern Hemisphere stratospheric polar vortex."**

**Jono Conway**

We thank Zachary Lawrence for his thoughtful and positive comments on our manuscript. Below we provide responses to his specific comments:

Comments and Questions

1. The introduction is rather short. While it is nice to have it be concise, the introduction in its present form does not discuss any previous examinations of the SH vortex boundary region. I do not know of any papers that show or mention the bifurcation of PV gradients, but there are many studies (some of which are already cited in the paper) that have discussed and shown that the SH vortex boundary region is often quite wide, especially relative to the NH vortex. Many of these studies have used more complicated diagnostics than PV gradients (e.g., effective diffusivity) to show this, so mentioning and briefly discussing such papers would reinforce the following statement in the intro "Because of the minimal computational requirements, the PV gradient is an attractive method to define the vortex boundary region." Some potential relevant references are already cited in the paper; for example, see figures and related discussion on: Figures 3 – 5 in Paparella (1997) show trajectories of balloons trapped inside the SH polar vortex core and polar vortex boundary; Figures 1 & 2 in Lee et al. (2001) show the width of the vortex boundary region in effective diffusivity during 1996; and plates 1 – 4 in Haynes and Shuckburgh (2000) show monthly means of effective diffusivity that can allow comparison of the widths of the Arctic vs Antarctic vortex boundaries. Other more recent studies that may or may not be relevant to look at include (but certainly are not limited to): de la Cámara et al., JAS, 2012; Abalos et al., QJRMS, 2016; and Curbelo et al., NPG, 2017.

We now make a larger discussion of the various methods to detect the vortex boundary in the introduction.

2. Tied to 1, the results shown in the current paper could be made more impactful in both the introduction and conclusions by being more clear about the "magnitude" of what is shown. In my opinion, the paper sells itself a bit short. More specifically, I mean that many previous studies (including some of the ones discussed in comment #1) only show results relevant for a single dataset for one to a few winters; this paper shows results relevant for two reanalyses for years from 1979 to roughly the present. I think that is worthy of being highlighted as one of the strong points of the paper.

We have highlighted these points in the conclusions.

3. This is very minor, but in section 2.1, the years used for ERA-Interim and CFSR should be listed.

Dates for each reanalysis have been added.

4. Figure 1 and Table 1 are helpful, but it would be nice if a new figure was included showing EqL profiles of PV gradients for different values of the BI. For example, a 2 x 3 figure with panels showing (along with dates, isentropic levels, peak ratios, and dip fractions listed) representative cases with BI = 0, 2, 4, 6, 8, and 10. I think this would give the reader a better sense of the connection between BI and the geometry of the PV gradient profiles, as well as enhance understanding and the significance of the other figures in the paper.

During the preparation of the manuscript we considered adding a figure like this, but opted not to for brevity. We describe the range of bifurcated profiles in paragraph 3 of Section 2.2.

5. In Section 3.1 and Figure 2, is it really accurate to say that there are two regions of enhanced PV gradients? Since there is no clear dip, it looks as if there is only one broad region of enhanced gradients in all cases, even though the location of the climatological maximum gradients moves "equatorward" with height in August & September. This pattern seems to be clearly formed from averaging many cases with bifurcated PV gradients having peaks in different EqL locations, which would strengthen the idea that the PV gradient bifurcation needs to be examined on a year-to-year basis to really see and understand it (i.e., as is shown in Figure 3).

Agreed, the two regions are most clearly identified in the individual years, whereas the climatological gradients smear the gradients. We have reworded this sentence:

"In August and September, PV gradients intensify in concert with the region of strong wind speed becoming predominately confined to latitudes poleward of -50°. While increased climatological PV gradients occur over a broad region, at lower levels these are nearer the poleward edge of the jet, while at higher levels these are nearer the equatorward edge of the wind speed jet (Figure 2 c,d)."

6. In Figure 3, I would be curious to see whether the bifurcated structure also shows up in "regular" zonal means of meridional PV gradients (i.e., with respect to "regular" latitude), since the SH flow and PV distributions are usually relatively close to zonally symmetric. If the bifurcated structure also exists in the zonal mean picture, this could be an interesting result with further implications (e.g., for wave guiding/propagation).

Agreed, this would be an interesting analysis, but it is beyond the scope of the paper so do not plan to make these analyses here.

7. The comparison with CFSR in section 4.1 is so short that it almost seems unnecessary. First of all, why not show all the same panels for CFSR in Figures 7 and 8 as those in Figures 4 and 5? And since section 4.1 is so short, why not fold the CFSR results from Figures 7 and 8 into Figures 4 and 5 (and maybe even Figure 6) and discuss the comparisons in sections 3.3 and 3.4 directly? Figures 4 and 5 would obviously increase in size (and more labels would be necessary), but there would be an advantage to having the results from the two reanalyses side-by-side for easier comparison, and less figures overall. This would also reinforce one of the strengths of the paper (see my comment #2).

Because of the better vertical resolution of the ERA-Interim data set, we prefer to restrict the main results to ERA-Interim and keep the comparison with CFSR separate. The comparison needed, but simply there to show that the patterns we observe are real phenomena, and not the product of the methods used to calculate them in a particular reanalysis. Because the ERA-Interim and CFSR data sets are produced on different potential temperature levels, a formal quantitative comparison is not possible. We have added additional panels to Figures 7 and 8 to present further results for CFSR

8. Figure 9 and its discussion in Section 4.2 would be a bit more useful/significant if Figure 9 showed composites of some form, so that readers would know the discussion is more broadly relevant than to a single day of data. I do realize, however, that this may not be easy or possible to do since the locations of peaks in PV gradients (and hence the geometry of the PV and relative/absolute vorticity profiles) will vary from day to day – but it is something that might be worth a try.

We have also provided a new figure (new Figure 10) showing height-latitude composites of vorticity and static stability gradients, averaged from the 5 years with the highest BI at mid-levels. When PV gradients are bifurcated at mid-levels during August and September (Figure 10 d-i), there are distinct

locations for the primary and secondary peaks and the composite analysis reveals some useful patterns. The composite analysis reinforces the findings from the case study day, showing BI is associated with elevated vorticity gradients in two locations, along with increased static stability gradients at the inner location.

9. Also in Figure 9 – how were the fields of relative/absolute vorticity obtained? Were they derived from the potential vorticity and temperature fields, or the wind components, or were they downloaded?

Relative vorticity was downloaded from the ECMWF online archive. Absolute vorticity was calculated from relative vorticity and latitude at each grid point. The lapse rate of potential temperature was back-calculated from PV, relative vorticity and latitude at each grid point and the equation for Ertel PV. These details have been added to the text.

10. The following is just an idea I had, but something the authors might consider: The current organization of the paper is fine, but since the paper is largely focused on presenting the methods and diagnostics, there is an alternative organization I see that might flow better for readers (and would only require minimal work). Arguably the bifurcation detection methods and diagnostics are part of the results of the paper. Thus, section 2 could be changed to a "Data and Background" section wherein section 2.1 would remain the same, section 2.2 would discuss how the EqL profiles of PV and PV gradients are obtained, but a new section 2.3 would be inserted (the background) that included Figures 2 and 3 (which would become Figures 1 and 2) and their discussion. The BI panel in Figure 3 would have to be changed to something like "average peak separation in degrees EqL" since BI would not be introduced yet. Then in section 3, the description of the bifurcation detection and diagnostics (and the current Figure 1) would be included in section 3.1 with a title like "PV gradient bifurcation detection and diagnostics." This organization would lead readers from the climatologies (which smear out the bifurcated gradients), to a single SH winter that illustrates the phenomenon, to the detection methods and bifurcation diagnostics, and finally to the results/figures that directly show the BI and other diagnostics (which, with this organization, would have just been defined). Again, this is just an idea; the authors should feel free to flatly reject it!

Thanks for the reorganisation suggestion – but we've opted to keep our original organisation.

**Response to reviewer comment by Petr Šácha on "Bifurcation of potential vorticity gradients across the Southern Hemisphere stratospheric polar vortex."**

**Jono Conway**

I think that it is absolutely vital for your study to reconsider your results with respect to the recently published work by Serra et al. (2017). Their geodesic method identifies a materially optimal vortex boundary. More to that, Serra et al. (2017) show also a comparison of their method with a "Nash" vortex boundary. To cite some of their conclusions: "Nash method is frame dependent and nonmaterial, hence a priori unsuitable for a self-consistent detection of coherent transport barriers." "Given its Eulerian (nonmaterial) nature, the Nash edge evolves discontinuously, with visible jumps in position and shape over time."

My personal opinion would be that your analysis of PV gradients may be untouched, but the relationship to mixing barriers needs to tackle these new findings.

We thank Petr Šácha for bringing to our attention the recent paper published by Serra et al 2017 'Uncovering the Edge of the Polar Vortex'. This paper presents a method for detecting the area of the relatively undisturbed airmass within the polar vortex using a Lagrangian framework and thresholds in deformation coefficients along the boundaries. They provide a brief comparison of their kinematic boundary to that defined using the method of Nash et al (1997) for the Northern Hemisphere polar vortex for a short period 2013.

As mentioned in the comment and the response by Gloria Manney, the findings of Serra et al. 2017 are mostly relevant to a definition of the the vortex edge, and not to the main result of our manuscript (which concerns PV gradients). We agree there is a need to thoroughly review the definitions of the vortex boundary, formally compare the various methods to detect these boundaries and quantitatively compare the transport and mixing properties associated with the vortex in the Southern Hemisphere. But this task is well beyond the scope of the current manuscript. We have included Serra et al. (2017) in list of Lagrangian metrics used to identify the vortex edge in the introduction.

[revised manuscript text omitted]